# Counter-Mapping in Geographic Education: A Novel Approach to Understanding Urban and Cultural Dynamics in Cities

Seila Soler [1,*] and Pablo Rosser [2]

1. Faculty of Humanities and Social Sciences, Isabel I University, 09003 Burgos, Spain
2. Faculty of Education, International University of La Rioja, 26006 Logroño, Spain; pablo.rosser@unir.net
* Correspondence: seilaaixa.soler@ui1.es

**Abstract:** This study assessed the impact of counter-mapping on university students' perception and knowledge of their cities, focusing on cultural diversity, urban dynamics, and social challenges. Using a quasi-experimental design with pretests and posttests in the province of Alicante, Spain, changes in the understanding of urban geography, everyday life, and cultural diversity were analyzed among 54 Geography Didactics students. Likert scale surveys and open-ended questions were employed, complemented by statistical and qualitative analyses, to measure knowledge and perceptions before and after the counter-mapping project. The results indicate that although quantitative correlations did not show statistically significant significance, qualitative analysis revealed significant and profound learning. Participants uncovered the hidden layers of their urban environments and gained a better understanding of the complexities and challenges of their cities. The project promoted the development of critical skills such as analytical thinking, active research, and effective communication, proving to be a valuable pedagogical tool that surpasses the limitations of traditional teaching methods and encourages active and committed citizenship. This inclusive and multidimensional approach enriches the diversity of experiences and knowledge, significantly contributing to the teaching of urban subjects, community engagement, and social responsibility and laying the groundwork for the proper tourist, cultural, social, or urban planning of city spaces.

**Keywords:** counter-mapping; urban perception; geographic education; touristic planning; critical skills



## 1. Introduction

In the context of urban transformation towards sustainability and the valuation of cultural heritage, the relevance of exploring innovative educational methodologies that promote a deep understanding of the urban and cultural environment is underscored. Counter-mapping, emerging as a powerfully innovative pedagogical tool within geographical education, offers novel perspectives for understanding and reimagining the urban and cultural environment, thus challenging traditional mapping methods that often reflect a top-down perspective and are planner-centered [1,2]. This approach facilitates a richer understanding of urban dynamics from a bottom-up perspective, valuing the perceptions and experiences of local residents, which is crucial for promoting the development of critical skills and an active and committed citizenship.

Faced with the challenge of innovating within the academic realm, this study employs a single-case approach not as a limitation but as a deliberate strategy for delving deeper into the specific urban and cultural dynamics of Alicante. This method allows not only for a detailed and contextualized understanding but also for the exploration of innovative solutions to complex problems in urban planning and geographic education. By focusing on a specific case, our research uncovers previously undocumented layers and dynamics, thereby offering new perspectives that challenge traditional understandings and enrich the academic dialogue on sustainable development and social inclusion in urban contexts. This

approach reflects a commitment to generating applicable and transformative knowledge, aligned with contemporary demands for innovation and academic relevance.

The selection of counter-mapping as the core of our educational project is justified by its capacity to foster critical thinking among students about the challenges and opportunities their cities face. This aligns with the principles of critical pedagogy, which seeks to challenge educational inequalities and promote social justice, emphasizing the importance of an education that cultivates critical awareness and encourages transformative action [3,4]. By integrating counter-mapping within this pedagogical framework, our study aims to explore its potential to encourage among students a critical reflection on how they can contribute to the creation of more inclusive and appealing urban environments for both residents and visitors.

This methodological approach responds to the urgent need to adopt sustainable and responsible practices in tourism and urban development, where the preservation of cultural heritage and the inclusion of diverse voices in the planning process are indispensable. Thus, our paper examines the role of counter-mapping not only as an educational strategy but also as a means to rethink and redesign cities in a way that aligns with the goals of sustainability and responsible tourism [5,6]. Critical thinking, based on our research and project, is a pedagogical approach that has gained popularity worldwide in the past decades. However, its application in education has not been without its problems and challenges [7–9].

Active methodology and critical pedagogy stand out in contemporary education for their focus on practical experience and the construction of knowledge relevant to the social reality and everyday life of students [10]. Reis and Formosinho highlight their benefits for creativity, active participation, and knowledge retention through techniques such as project-based learning and the flipped classroom [11]. However, critical pedagogy is distinguished by its commitment to addressing social inequalities and promoting social justice, while active methodology focuses more on the learning process and student participation [1,3,12,13]. According to Shor, critical pedagogy seeks to transform power structures within and outside the classroom, emphasizing that education must be contextualized to face the specific realities and challenges of each environment, especially in urban and multicultural contexts [14–17].

Approaching the more specific premises that have inspired our project, it is important to bring in the contributions of Jean-François Lyotard to postmodernism—fundamentally, his emphasis on language games as essential forms of communication and expression in society [18]. Lyotard proposes that, in the postmodern era, knowledge has become decentralized, freed from the control of a central authority, implying a significant reconfiguration in the pedagogical realm [2,18–21]. This decentralization suggests that students should be seen not as mere passive recipients of information but as active participants in the generation and production of knowledge, contributing their unique perspectives and experiences to the learning process [22].

Moreover, Lyotard and Polanyi highlight how factors such as linguistic and cultural affinities influence the acquisition of scientific knowledge, underlining the importance of considering multiculturalism within cities and, therefore, in the classroom [23]. This approach proposes a paradigm shift towards a pedagogy that values and encourages diversity and inclusion, recognizing students as co-creators of knowledge within a broader and more diverse educational context.

Our educational project is also inspired by the theories of Henri Lefebvre, who emphasized collective and transformative action in education and other areas to modify social and political structures [24]. Lefebvre critically examines urbanization, applying semiotic, structuralist, and poststructuralist methodologies to analyze urban complexity and promoting a deep understanding of urban space [25,26]. He opposes modernist views, emphasizing lived experience and the creation of an urban utopia based on self-determination and the authenticity of social relationships and arguing that space is both a product and a medium of power relations [5,26–33].

Similarly, our educational approach is based on the principles of Carl Rogers, who highlighted the potential for personal growth and self-realization, promoting a learning environment characterized by empathy, congruence, and active listening. This approach encourages active student participation in their learning process, considering them unique and autonomous beings [34–38]. Rogers, a pioneer of humanistic psychology, sees education as a means for personal development and the construction of meaningful knowledge through self-directed learning. His methodology aligns with critical pedagogy, emphasizing critical reflection and social transformation in education [39–41].

Reflecting on post-truth in the context of critical pedagogy, we will highlight its impact on education, marked by an era where emotions and personal beliefs predominate over objective facts, complicating the educational field with the spread of fake news and information manipulation [42,43]. Frankfurt distinguishes deliberate deception from foolishness, with the latter being an indifference towards truth more harmful to the valuation of truthfulness [44,45]. Post-truth challenges the development of critical skills and evidence-based decision-making, encouraging reliance on unreliable sources [46–48].

Neil Postman warned about the risks of an educational environment dominated by media entertainment, limiting the depth of learning [49,50]. He suggests focusing on critical thinking and media literacy to counter these effects, with Bandura emphasizing the importance of appropriate models for social learning in post-truth environments [51,52]. Giroux investigates how media culture and commercialization affect education, promoting a critical education that resists manipulation and misinformation [6,53–56].

Not forgetting that our project carries a significant symbolic load, it is important to highlight Clifford Geertz's concept of "symbolic efficacy", referring to the power of cultural symbols in human perception and action, highlighting the influence of symbols in the construction of meaning and their relevance in education [57–60]. In this sense, we focus on edusemiotics, a pedagogical approach centered on meaning and communication that examines how signs and symbolic systems affect knowledge and learning, emphasizing the use of visual resources, interactive practices, and technologies to facilitate the construction of meanings [61–64]. This approach includes the analysis of signs in the educational environment, the consideration of the cultural and social context in education, and the promotion of meaning construction by students [65–67].

But Ford, advancing the ideas of Henri Lefebvre, believes that we can go beyond critical pedagogy to address these issues, exploring the connection between education and spatial production and proposing a distinct political pedagogy [5,44]. Ford criticizes the individualistic tendency of critical pedagogy, promoting the integration of education and critical geography to analyze and transform spatial and power structures in education [44,48]. This pedagogy seeks to reform social and spatial structures influenced by capitalism, considering education as a transformative force [46,47,68–70].

Insofar as critical geographers point out that cartography is always a political process situated in a social context with specific purposes and effects [71–76], we will use counter-mapping for this. Indeed, in recent years, there have been publications that have explored fundamental changes in the way cartography is understood and practiced as a social process [77]. Several authors have contributed to this discussion [72,75,78].

To conceptualize the use of counter-mapping, we have been guided by the postulates of Nelson & Chen [79], who present a problem-posing model based on Freire's methodology [1], structured in five phases to enrich teaching through critical approaches. This process begins with the exploration of relevant themes through critical dialogue, followed by the coding and decoding of these themes into symbolic representations. Subsequently, a reflective dialogue on the themes and their relationship with the students' experiences is promoted, leading to transformative actions in their context and concluding with a critical evaluation of the entire process. This model encourages active participation, critical reflection, and informed decision-making, seeking not only learning but also the development of social awareness among students.

## 2. Materials and Methods

This research analyzed the impact of a counter-mapping project carried out in various cities within the province of Alicante (Spain) by university students of the Didactics of Geography course in the Education degree program. Likert scale surveys and open-ended questions were used to measure the level of knowledge before and after implementing the project.

### 2.1. Objectives

#### 2.1.1. General Objectives

- Assess the impact of the counter-mapping project on university students' knowledge and perception of their cities, focusing on urban dynamics, cultural diversity, and social challenges.
- Examine how counter-mapping, as a pedagogical tool, can enrich the understanding of urban spaces and foster active and committed citizenship among students.
- Demonstrate the effectiveness of counter-mapping in developing critical skills, such as analytical thinking, active research, and effective communication, which are fundamental in the current urban and globalized context.
- Explore the possibilities of counter-mapping to overcome the limitations of traditional teaching methods, promoting multidimensional learning that integrates active participation, on-site exploration, and critical reflection.
- Contribute to the improvement of teaching related to urban life, community engagement, and social responsibility through the implementation and evaluation of the counter-mapping project.

#### 2.1.2. Specific Objectives

- Measure the change in the level of knowledge and perception about urban geography, cultural history, social diversity, and urban challenges among students before and after participation in the counter-mapping project.
- Analyze the transformation in the appreciation and understanding of the "invisible city" by students, highlighting the importance of personal and community narratives in the construction of urban identity.
- Evaluate the effect of counter-mapping in promoting critical reflection on urban spaces and in fostering a substantial change in the perception and appreciation of the cultural and social diversity of their cities.
- Identify how the counter-mapping methodology influences the development of critical thinking and creativity skills, essential for informed and active citizen participation.
- Investigate the relationship between students' prior knowledge of their cities and the learnings acquired during the project to determine the accessibility and universality of counter-mapping as an educational tool.

### 2.2. Study Design

A quasi-experimental design was adopted, with pretest and posttest measurements, to evaluate changes in students' knowledge and perceptions of urban geography, everyday life, and cultural diversity after participating in the counter-mapping project.

### 2.3. Surveyed Population

The surveyed population consisted of 54 students enrolled in the Didactics of Geography course within the Education program during the academic semester spanning from September 2023 to January 2024. The anonymity of all participants was ensured, with the information collected being treated with the utmost confidentiality and used solely for academic and research purposes, in compliance with ethical research standards.

*2.4. Data Collection Instruments*

- Likert Scale Surveys: Measured perceptions and attitudes towards city geography, everyday life, cultural diversity, and citizen participation before and after the project.
- Open-Ended Questions: Provided detailed information on students' experiences and perceptions regarding the counter-mapping project.

*2.5. Data Analysis*

Various statistical and qualitative analyses were conducted:

- Descriptive Analysis: The mean, standard deviation, and variance were calculated for each variable.
- *T*-Test for Repeated Measures: Compare the level of knowledge before and after the project.
- Wilcoxon Test for Related Samples: Evaluated changes in city knowledge and socio-emotional perceptions.
- Regression Analysis: Determined the relationships between knowledge and perceptions pre- and post-project, including the impact on historical and cultural knowledge, the perception of everyday life, and the "invisible city".

*2.6. Enhancing Educational Assessment through Integrated Methodologies*

Our evaluation framework ensures a coherent integration of qualitative and quantitative methods in order to strengthen the correlation between these methods and our instructional design. This objective focuses on two main strategies. On one hand, in the evaluation, we have ensured that both the qualitative and quantitative methods are aligned with each other and directly linked to the learning objectives and instructional strategies of the course. This alignment enhances the accuracy in measuring the impact of pedagogical innovations. As Greene et al. emphasize [80], the combination of qualitative and quantitative methodologies allows for the cross-verification of data and facilitates the generation of richer, more contextual ideas, which is crucial in our educational context.

On the other hand, we have ensured the coherence and validation of our findings by adopting a triangulation method, following the recommendations of Krippendorff [81]. This approach utilizes multiple data sources and methods to ensure a more comprehensive and reliable assessment of educational impact. For example, quantitative data from pretests and posttests provide quantifiable indicators of knowledge acquisition, while qualitative data, derived from student reflections and interviews, offer deeper insights into their understanding and practical application of knowledge.

The quantitative analysis of survey data provides a general measure of effectiveness, while qualitative perceptions from focus groups clarify the underlying reasons for the trends observed in the quantitative data.

This robust and well-founded methodological approach ensures that the assessment of teaching effectiveness is comprehensive, reliable, and closely aligned with our pedagogical objectives. These methodological adjustments respond to and reflect the innovative nature of our instructional design, drawing on the relevance of qualitative methodology to interpret underlying meanings and contexts in educational data, as highlighted by Drisko and Maschi in their discussion on the importance of qualitative content analysis in education [82].

*2.7. Perception Evaluation*

Changes in the perception of cultural diversity, citizen participation, commitment to personal and institutional actions, and the level of creativity and critical thinking were examined. Quantitative and qualitative analyses were used to provide a comprehensive view of the project's effects. The Atlas.ti software (7.5.18) (Berlin, BE, Germany) was utilized for qualitative analysis, while SPSS Statistics (28.0.1.1 (14)) was employed for quantitative analysis.

*2.8. Counter-Mapping Method*

To implement a practice using critical cartography and mapping, taking into account the 4Ks [83] and the problem-posing phases of Freire and Nelson & Chen [79], the following steps were followed:

- Awareness: An activity was proposed that allowed students to become aware of the social reality not represented in official cartographies.
- Research: Subsequently, students were asked to investigate the social realities they had identified in the previous activity. For example, students conducted interviews with residents of places not appearing on official maps to understand their stories and needs.
- Critical Dialogue: Once students had gathered information about social realities, a critical dialogue was initiated in which the official representations of urban space were questioned, and reflections on the possibilities for transforming urban space were encouraged.
- Transformative Action: Finally, a transformative action was proposed in which students used critical cartography and mapping to make visible the social realities they had identified and to propose solutions to the needs of the inhabitants of places not appearing in official cartographies.

The Learning Stations carried out in the aforementioned phases were as follows:

- The Historical City: The urban space and its transformation throughout history were analyzed, delving into what remained of that history, what those neighborhoods were like, and how people lived in them, sometimes coexisting with tourism or, in other cases, with poor building conditions, a lack of accessibility or services, etc.
- The Lived City: In this case, the everyday life of different population groups (children, youth, adults, and the elderly), leisure and available culture, consumption and services, prices of things, tourism, etc., were analyzed.
- Invisible Cities: This is the most abstract aspect, due to its invisibility, but it is usually the most impactful and real of all when analyzed. They are the spaces and people that institutions do not want to be seen, which are isolated or marginalized.
- City Among Cities: This is also apparently difficult to appreciate, but, upon analysis, it was seen whether some environments or neighborhoods preserved traditions or identity markers more than others, which had their characteristics beyond belonging to the wider city space. These are places where the "official" culture was not followed but rather their own. Their own spaces were neither better nor worse but where things were done that are not experienced in other areas.
- "Congress" of Results: After completing the work in all its phases, the counter-map was graphically made and presented by each group to the rest of the class, explaining its main characteristics and opening a dialogue among the entire class to search for contradictions, invisibilities etc. and also compare them with the rest of the projects carried out by other students.

All of this was preceded by responses to the pretest on critical thinking pedagogy, potential tools for classroom use, and feelings before tackling the project, based on a Google form about cognition, emotion, and satisfaction. A similar survey was passed at the end of the project.

*2.9. Pedagogical Design and Methodology*

This semester-long course has been meticulously designed to integrate the counter-mapping method as its central pillar, adapting the sequential applicability of content from Bauer's methodology [84] through five dynamic and participative learning stations. Each station enables students to investigate and redefine their perception of urban space through a critical and participative approach, facilitating deep and applied learning of urban and cultural dynamics. In this respect, the Learning Stations approach offers a pedagogical methodology that allows for working in different thematic areas simultane-

ously, resulting in a more comprehensive and multidisciplinary learning experience for students, enhancing cognitive achievement and social, practical, and professional competencies [84,85]. Moreover, this approach promotes active student participation in their own learning process, enabling them to develop critical and creative skills to address a complex and relevant problem for our society, with the role of the faculty being to accompany them in this process [86,87].

- Station 1—The Historic City: Students employ counter-mapping to explore the hidden layers of urban history, discovering how the past has shaped the current configuration of the city and continues to influence its future developments.
- Station 2—The Lived City: In this phase, students map the everyday life of the inhabitants, interacting directly with local residents to understand how people and urban spaces interact in continuous symbiosis.
- Station 3—The Shared City: Focusing on public spaces, this station challenges students to map and analyze social interactions and conflicts, offering a view of community cohesion and tension.
- Station 4—The Invisible City: Students identify and represent often invisible elements of the city, such as marginalized communities and subcultures, highlighting the hidden diversity within the urban fabric.
- Station 5—Cities within the City: This final station invites students to discover and map the coexistence of multiple "cities" within a metropolis, each with its own challenges and unique characteristics.

Each learning station is designed to be autonomous, allowing students to work in small teams to research, discuss, and create cartographic representations that reflect their discoveries and analysis. This pedagogical approach not only enriches the students' geographical and cultural understanding but also fosters critical thinking skills, active research, and effective communication. The curriculum includes assessments and practical projects that culminate in a "Results Congress", where students present their maps and analyses, fostering critical and reflective dialogue on the urban dynamics uncovered. This innovative pedagogical design ensures that students are prepared to apply their knowledge in real-world contexts and make significant contributions to urban planning and development.

*2.10. Engaging with Urban Dynamics and Cultural Diversity through Counter-Mapping*

Counter-mapping in our study involves a series of structured activities that engage students in a deep, reflective interaction with their urban environment. This process begins with the identification of culturally and socially significant yet often overlooked urban areas. Students, guided by instructors, use various data collection methods including direct observations, interviews with local residents, and the review of secondary sources to gather rich, qualitative data. Practically, the implementation of counter-mapping is carried out through a series of workshops where students learn to create maps that reflect both visible and invisible urban elements. These maps highlight differences in cultural practices, economic conditions, and social interactions across different urban areas.

The counter-mapping process actively involves students in analyzing how urban areas evolve over time due to factors like migration, economic changes, and policy shifts. This dynamic view helps students appreciate the fluid nature of urban spaces and the interplay of various cultural and social forces shaping these areas. By focusing on diverse neighborhoods, students explore how different communities contribute to the vibrancy and complexity of urban life.

Cultural diversity is at the heart of our counter-mapping methodology. Students are encouraged to explore and document diverse cultural expressions found within urban settings—from street art and public performances to local festivals and community gatherings. This exploration is critical for understanding the cultural richness and varied human experiences that constitute urban centers.

The methodology fosters critical thinking by challenging students to question conventional representations of urban areas and to consider the biases that might influence these

depictions. Creativity is stimulated through the design of unique map layouts that depict the students' interpretations and insights regarding urban diversity and dynamics. The creative process is supported by group discussions, which help refine their outputs and deepen their understanding of the subject matter.

By integrating this expanded methodology into our curriculum, we aim to enhance students' ability to think analytically and creatively about urban geography. The hands-on experience with counter-mapping equips them with the skills to conduct independent research and contribute thoughtfully to discussions on urban planning and cultural preservation (Figure 1).

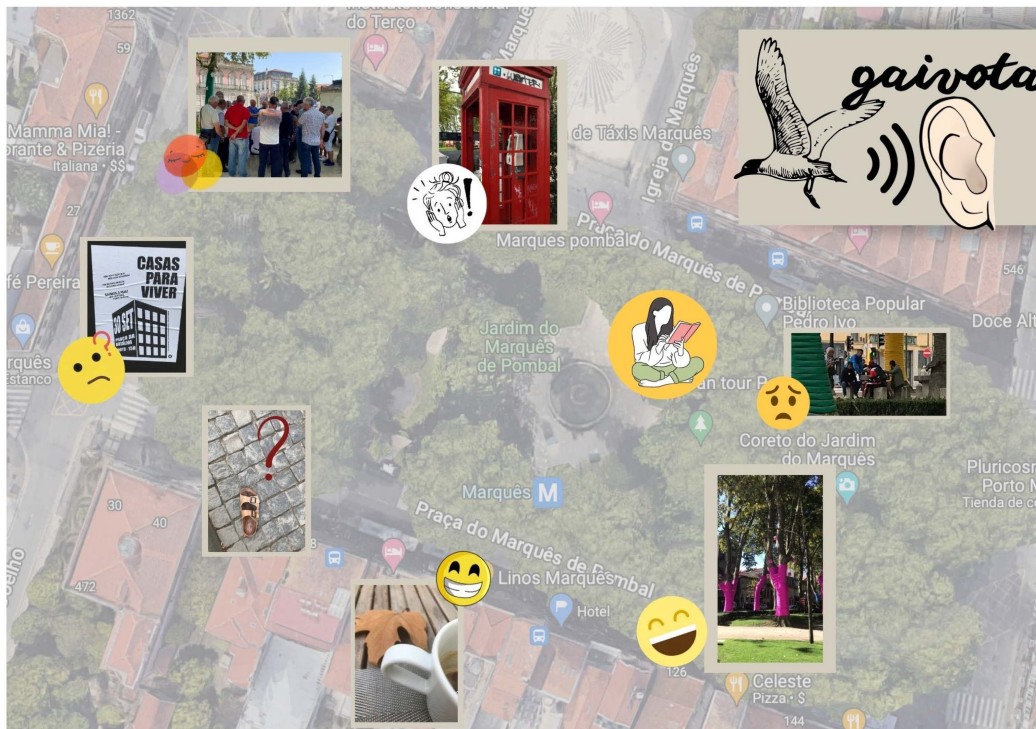

**Figure 1.** Example of counter-mapping conducted by the authors in a neighborhood of Porto (Portugal). Source: Authors' elaboration.

## 3. Results

### 3.1. Descriptive Analysis

In the descriptive analysis of variables related to the counter-mapping project, aspects such as the number of responses, range, minimum and maximum values, total sum, and average were evaluated to understand the distribution and central tendencies of the participants' responses. The research showed that knowledge about the city before the project was average, suggesting a moderate understanding among the participants. This knowledge experienced a significant increase after the project, as reflected by the average changes in knowledge and the post-project knowledge level (Figure 2).

The data obtained from the counter-mapping project demonstrate a significant positive effect on the participants, evidenced by the increase in the averages of almost all the evaluated variables after participating in the project. This effect is observed in areas such as general knowledge and perception of the city and creativity and critical thinking, indicating that the project has been crucial in enriching the participants' awareness and understanding of their urban and cultural environment.

The data reflect variations in standard deviations, indicating different degrees of uniformity in the participants' responses. Variations in the standard deviation, both in decreases and increases, suggest that, while some aspects promoted greater alignment in

perceptions, others observed a diversity of opinions and experiences that enriched the analysis of the project's impact.

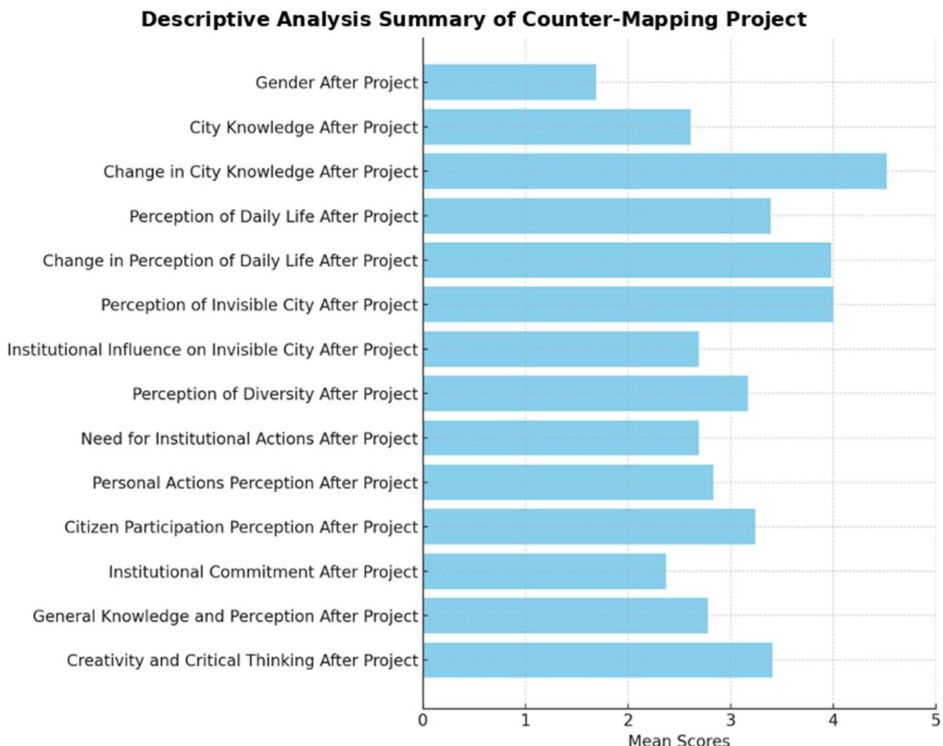

**Figure 2.** Descriptive analysis of knowledge variables after the project. Source: Authors' elaboration.

### 3.2. T-Test before and after Counter-Mapping

The application of the *T*-test for paired samples in the context of the counter-mapping project revealed significant changes in the participants' perceptions and knowledge before and after their involvement in the project (Figure 3). These changes reflect the project's influence on different dimensions of the participants' understanding and urban perception.

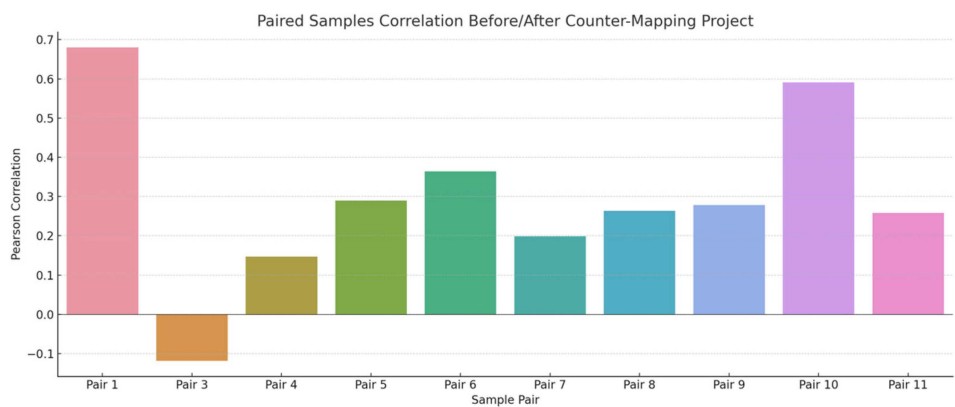

**Figure 3.** Pearson correlation for each pair of samples before and after the counter-mapping project. Source: Authors' elaboration. A horizontal line at y = 0 helps to differentiate between positive and negative correlations.

Significant increases in the level of knowledge about the city after the project (Pair 1) were identified, evidencing the project's effectiveness in enriching the urban and heritage knowledge of the participants. Additionally, notable changes were recorded in the perception of everyday life and in how information about the "invisible" city is understood

(Pair 3), indicating an increase in critical awareness of how information about the urban environment is presented and hidden.

The perception of cultural diversity also experienced a significant improvement (Pair 4), underlining the importance of recognizing the complexity and cultural richness of cities. From a transformative action perspective, both at the institutional (Pair 5) and personal levels (Pair 6), the project promoted the recognition of the need to engage in change initiatives, highlighting the relevance of institutional and personal participation and commitment to urban transformation. This recognition extended to the perception of the importance of citizen participation (Pair 7), reinforcing the idea that citizen involvement is fundamental for the development of sustainable and participatory cities.

The results also indicate a positive impact of the project on personal commitment towards urban change (Pair 8), emphasizing the crucial role individuals play in the sustainable transformation of cities (Pair 9). Thus, an overall increase in knowledge about the city (Pair 10) was observed, demonstrating the project's capacity to expand participants' urban understanding. Finally, the project had a favorable effect on promoting creativity and critical thinking (Pair 11), indispensable skills for addressing and solving contemporary urban challenges.

### 3.3. City Knowledge Level before and after the Wilcoxon Test for Related Samples

The Wilcoxon test, suitable for situations where differences between samples do not necessarily follow a normal distribution, provides a means to evaluate whether these changes are statistically significant without assuming the normality of distributions. Therefore, the application of the Wilcoxon test for related samples to analyze the level of knowledge about the city before and after the counter-mapping project offers a quantitative perspective on the educational impact of the initiative (Figure 4).

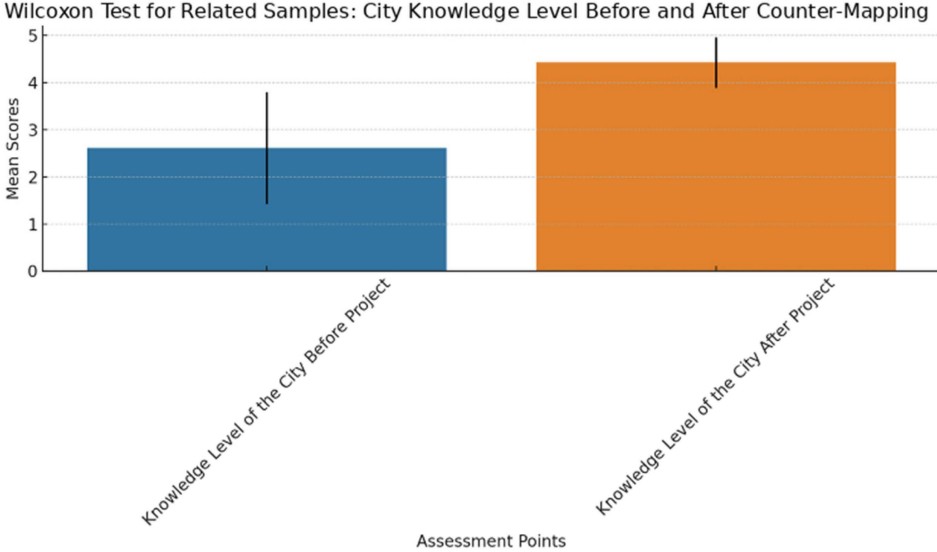

**Figure 4.** Means and standard deviations for two evaluation points: "Knowledge Level of the City Before Project" and "Knowledge Level of the City After Project". Source: Authors' elaboration.

With an initial mean of 2.6111 and a standard deviation of 1.18825 before the project, and a subsequent mean of 4.4259 with a standard deviation of 0.53560, the comparison of these data suggests a substantial improvement in the urban knowledge of the participants. The test result, with a W value of 0.0 and an approximate *p*-value of $2.00 \times 10^{-13}$, confirms that the difference in the level of knowledge about the city before and after the project is statistically significant at a conventional confidence level.

*3.4. Regarding Whether the Degree of Prior Knowledge Influences the Level of Knowledge Gained after the Project*

3.4.1. City Knowledge

As seen from the Wilcoxon test, there is a considerable increase in city knowledge after conducting the counter-mapping. But to what extent is this increase conditioned by prior knowledge? Investigating this link through Pearson's correlation and non-parametric correlations of Kendall Tau-b and Spearman Rho can be relevant and shed light on the learning dynamics underlying the project (Table 1).

**Table 1.** Correlations on city knowledge before and after the project. Source: Authors' elaboration.

| | | Historical and Cultural Knowledge Level before the Project | Historical and Cultural Knowledge Level after the Project |
|---|---|---|---|
| Historical and Cultural Knowledge Level Before the Project | Pearson Correlation | 1 | 0.680 |
| | Sig. (two-tailed) | | <0.001 |
| | Sum of Squares and Cross-Products | 74,833 | 22,944 |
| | Covariance | 1412 | 0.433 |
| | N | 54 | 54 |
| Historical and Cultural Knowledge Level After the Project | Pearson Correlation | 0.680 | 1 |
| | Sig. (two-tailed) | <0.001 | |
| | Sum of Squares and Cross-Products | 22,944 | 15,204 |
| | Covariance | 0.433 | 0.287 |
| | N | 54 | 54 |
| | The correlation is significant at the 0.01 level (two-tailed). | | |

The significant and robust correlation between prior knowledge and post-project acquired knowledge, indicated by a Pearson coefficient of 0.680, suggests that those students who had more solid initial knowledge about the city tended to enrich and deepen their understandings throughout the project. This is reflected in the personal narratives of the students, where those with prior familiarity with the city, through residency or previous interest, describe a learning process that deepens and expands their existing knowledge, often in surprising areas they did not anticipate.

On the other hand, students who started the project with little or no knowledge about the city report a significant enrichment of their historical and cultural understanding, driven by research activities and direct interactions with the urban space and its inhabitants (Figure 5). This demonstrates that the counter-mapping project catalyzed discovery and exploration, allowing these students to build a knowledge base from scratch.

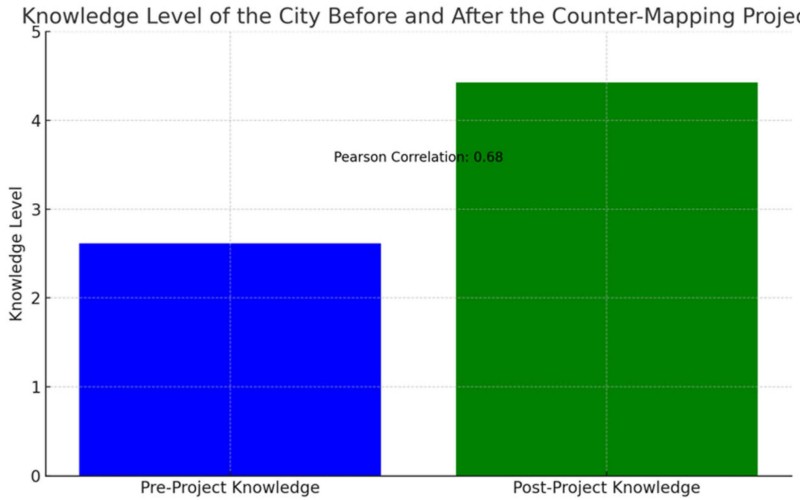

**Figure 5.** Correlation between the level of city knowledge before and after the project. Source: Authors' elaboration.

The experiences described by students from the qualitative analysis of open-ended questions (Table 2), such as interviewing city residents, exploring historical and cultural sites, and participating in research activities, illustrate how the learning process was influenced by a combination of fieldwork, community interaction, and personal reflection. This multifaceted approach seems to have fostered deeper and more contextualized learning, where the acquired knowledge is intimately linked to personal experiences and the perception of urban space.

**Table 2.** Qualitative Analysis. Open-ended questions on the variables "Level of historical and cultural knowledge before and after the project". Source: Authors' elaboration.

| |
|---|
| 2. What WAS your level of historical and cultural knowledge of that city before? Would you highlight anything specific that you knew before undertaking the project? |
| 5. Can you share some of the specific experiences or activities that contributed to your increased historical and cultural knowledge of your city AFTER your participation in the counter-mapping project? |

Source: Authors' elaboration.

### 3.4.2. Perception of Everyday Life

The low and non-significant correlation between perceptions of everyday life before and after the project, reflected in quantitative results, indicates that participation in counter-mapping has significantly altered the students' perceptions (Figure 6). This variability in responses and the lack of a strong correlation suggest that the project achieved its objective of changing perceptions on a specific subject—in this case, the everyday experience of urban life.

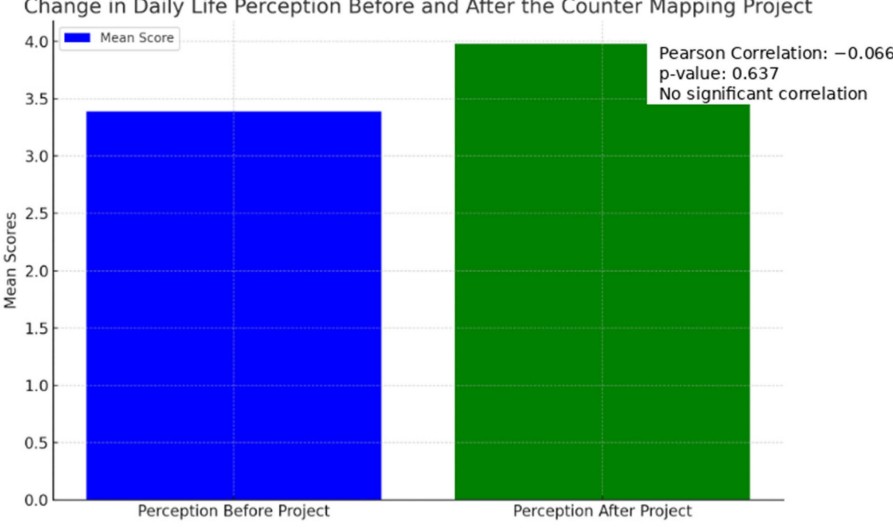

**Figure 6.** Correlation on the perception of everyday life before and after the project. Source: Authors' elaboration.

The diversity of individual experiences and perceptions highlighted in the qualitative analysis (Table 3) underscores how the project fostered a more nuanced and deepened understanding of the city, beyond previous conceptions based on assumptions or superficial knowledge.

**Table 3.** Qualitative Analysis. Open-ended questions on the variables "Perception of everyday life before and after the project".

| |
|---|
| 7. What was your perception BEFORE of everyday life in the city and its relationship with the tourist, economic, social, or cultural experience that you have highlighted from it? |
| 10. Can you share some of the specific experiences or activities of the Lived City that influenced your perception of everyday life in the city AFTER your participation in the counter-mapping project? |

Source: Authors' elaboration.

After the project, according to the qualitative analysis, a significant change in the students' perceptions was observed, reporting increased awareness and understanding of the complexity of everyday life in the city. This change is evidenced in the more critical evaluation of aspects such as the distribution and accessibility of urban services, the inclusion or exclusion of certain social groups, especially the elderly, and the identification of tensions between economic development and urban quality of life. Students highlight how the project allowed them to discover "the lived city", i.e., the everyday experience of its inhabitants, through direct interaction with the urban space and its residents. This discovery manifested in identifying social gathering areas, evaluating the supply and demand of leisure and cultural spaces, and reflecting on the dynamics of social inclusion and exclusion.

Indeed, counter-mapping revealed previously unnoticed aspects of everyday life in the city, showing how urban spaces are lived and experienced in different ways by their inhabitants. Through this methodology, students were able to identify and analyze daily routines, spaces of social interaction, and cultural dynamics that characterize the "Lived City". This approach not only enhanced their understanding of the diversity of urban experiences but also allowed them to reflect on the implications of these dynamics for urban planning and sustainable development. Through interviews and direct observation, the participants collected narratives that highlight the importance of considering the needs and preferences of all citizens in designing more inclusive and accessible cities.

### 3.4.3. Influence of the Information from the Invisible City

The low correlation between perceptions before and after the project, according to quantitative analyses, suggests that the project was effective in changing the participants' perceptions (Figure 7). This change is not reflected in a strong linear relationship between pre- and post-project perceptions, indicating that counter-mapping has provoked a significant reevaluation of how students understand official information and its impact on city perception.

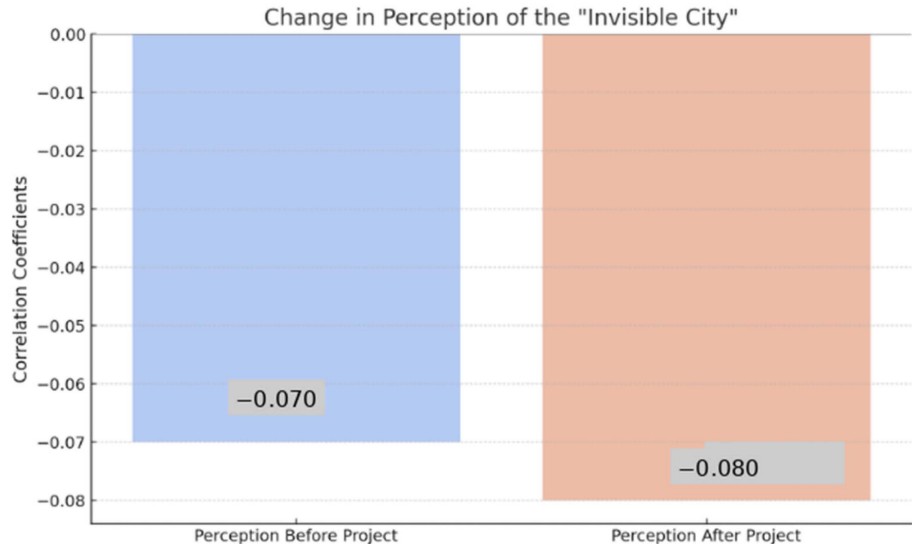

**Figure 7.** Correlation of the perception of the influence of information from the Invisible City before and after the project. Source: Authors' elaboration.

After participating in the counter-mapping project, students reported through qualitative analysis a greater awareness and understanding of the existence of an "invisible city" and the mechanisms by which official information may contribute to its concealment (Table 4). They identified specific aspects where official information has omitted or minimized certain urban realities, leading to a loss of city identity or the invisibility of certain

societal sectors. This change in perception reflects a deeper understanding of urban life complexities and the importance of considering all voices and experiences within the city.

**Table 4.** Qualitative Analysis. Open-ended questions on the variables "Perception of the concealment of the "invisible city" before and after the project".

| |
|---|
| 12. Were you aware BEFORE the counter-mapping that official information could be generating the existence of an "invisible city" in your environment? If so, can you provide examples or reasons that led you to this reflection (BEFORE counter-mapping)?<br>15. Can you share your reflections and observations after your participation in the research of the Invisible City? Have you identified any specific aspect in which official information has contributed to hiding an "invisible city", to the loss of city identity, or to the invisibility of certain sectors of society in the city? |

Source: Authors' elaboration.

Indeed, the counter-mapping project also addressed the concept of "invisible cities", those facets of the city that are often marginalized or excluded from the official urban discourse. Students explored and mapped these hidden "cities", allowing them to understand how official information can sometimes conceal significant and complex realities. Through the collection and analysis of unconventional data, counter-mapping facilitated a deeper understanding of the city's diverse layers, promoting a more complete and nuanced view of its social and cultural structure. This section of the project highlighted the importance of including marginalized perspectives in urban decision-making processes, pointing towards the construction of more equitable and representative cities.

3.4.4. Perception of Cultural Diversity before and after the Project

Through participation in the counter-mapping project, students began to recognize and value the richness of cultural and social diversity present in their environments (Table 5). They identified specific areas within their cities that operate with their dynamics, highlighting how these spaces offer services, traditions, and forms of social interaction that differ from other parts of the city. The project allowed them to discover "cities within a city", recognizing the importance of these micro-communities for the broader social and cultural fabric.

**Table 5.** Qualitative Analysis. Open-ended questions on the variables "Perception of cultural diversity before and after the project".

| |
|---|
| 17. What reflections did you have, BEFORE the counter-mapping project, about cultural and social diversity? Did you consider the possibility of there being "cities within a city"?<br>20. Can you share your reflections and observations after your participation in this research project? Have you identified any specific aspect in which the research influenced your perception of cultural and social diversity in the "cities within the city" or its importance for community development? |

Source: Authors' elaboration.

This transformation in the students' perceptions aligns with quantitative results that showed a moderate correlation in the perception of cultural diversity before and after the project (Figure 8). This moderate change suggests that initial perceptions were not completely displaced but were enriched and expanded, reflecting a learning process that adjusts and deepens previous understanding without completely discarding it.

Incorporating the theories of Freire and Lefebvre [1,5], the counter-mapping project has proven to be a transformative educational tool, not only for students of urban planning and geography but also as an inclusive mechanism for community participation. By fostering critical dialogue on cultural diversity, counter-mapping enables participants and the local community to rediscover and appreciate the richness of their environment, promoting a more informed and profound social engagement.

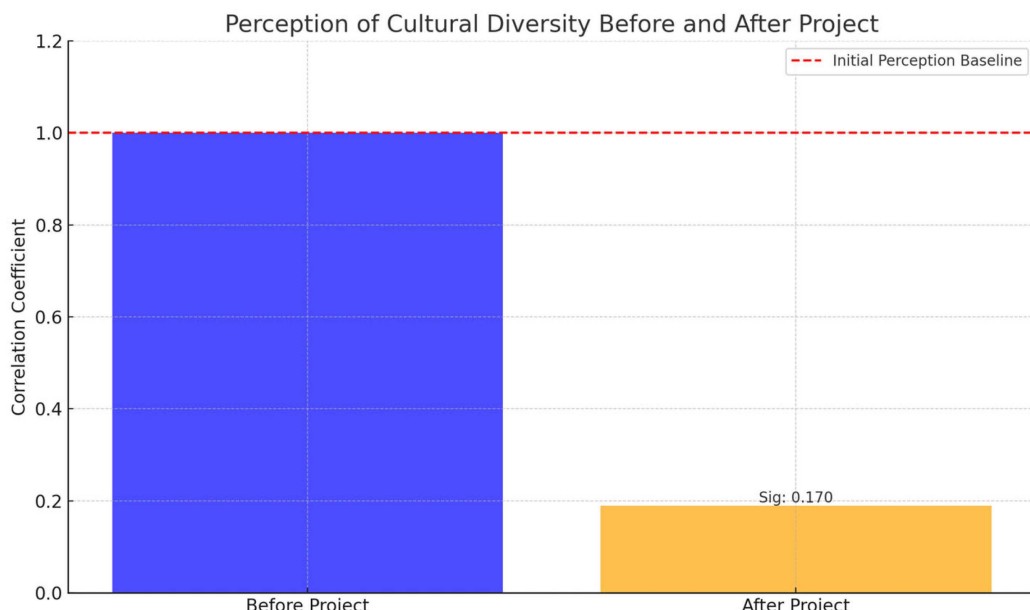

**Figure 8.** Correlation of the perception of cultural diversity before and after the project. Source: Authors' elaboration.

3.4.5. Perception between Transformative Action at the Institutional Level before the Project and Institutional Commitment after the Project

Before the project, awareness of specific needs for change in their communities varied, with some students pointing out the lack of adequate infrastructure, insufficient services, and social issues requiring attention. This variability in initial perception suggests that, while there was a general awareness of the need for change, the level of commitment and specific understanding of how to contribute to that change was limited.

Students' responses (Table 6) reflect a wide range of concerns, from the need to improve public transportation and cultural infrastructure to addressing vulnerable groups such as the elderly and including adequate leisure spaces for youth. The lack of knowledge or passivity regarding these issues before the project indicates that, although there was a perception of need, it did not necessarily translate into active action or commitment.

**Table 6.** Qualitative Analysis. Open-ended questions on the variables "Awareness of the need for institutional actions before and after the project".

| |
|---|
| 26. Provide specific examples (if any) of situations or experiences that influenced your perception of the need for transformative actions in your city BEFORE counter-mapping. |
| 27. Can you share examples of situations or events that led you, BEFORE counter-mapping, to consider your role in implementing transformative actions in your city? |

Source: Authors' elaboration.

The moderate correlation found quantitatively supports this evolution, suggesting that the project catalyzed greater engagement with institutional change, aligning with an improved perception of personal and collective capacity to influence urban dynamics (Figure 9).

Despite this positive evolution, the moderation of the correlation also points to a diversity in the change experience among participants, consistent with varied responses about their level of awareness and commitment before the project. This underscores the complexity of influencing perceptions of and attitudes towards institutions and the importance of differentiated strategies that consider individual and collective experiences to foster deeper and sustained engagement with transformative actions at the institutional level.

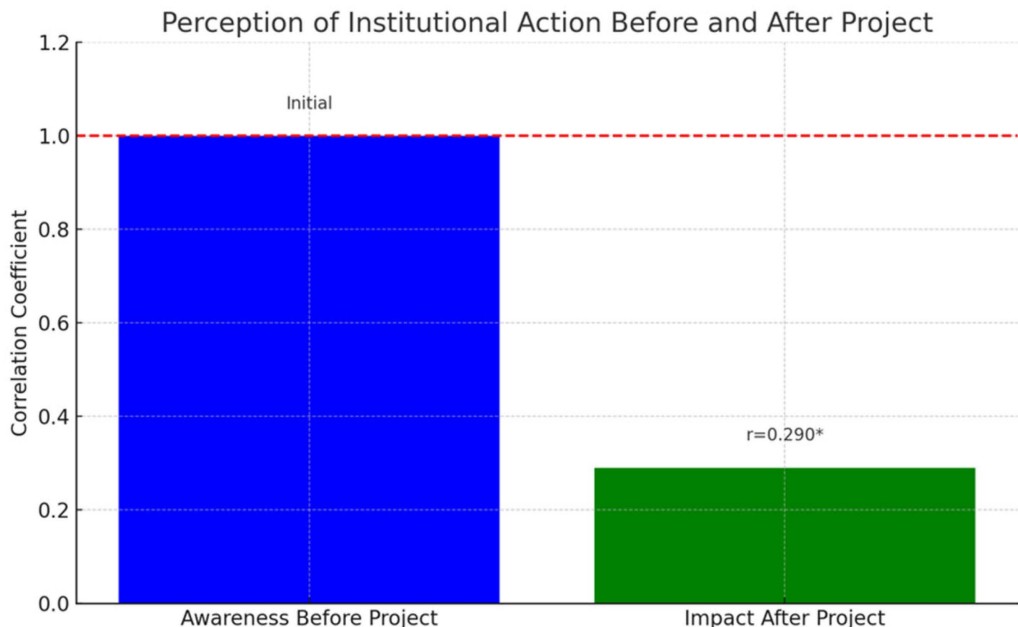

**Figure 9.** Correlation of the perception of institutional action before and after the project. Source: Authors' elaboration. The symbol * denotes statistically significant correlations.

The counter-mapping methodology aligns with Rogers' critical vision [37], transforming how students and other urban stakeholders, including NGOs and state institutions, perceive and relate to the "invisible cities". By revealing these hidden areas, counter-mapping acts as a bridge for social inclusion, highlighting the need for urban policies that consider all facets of city life.

3.4.6. Perception between Personal Transformative Action and Personal Commitment before and after

The moderate correlation underscores the significant impact that the initiative has had on participants (Figure 10). This link suggests that the activities and experiences proposed within the project framework have motivated participants to engage more deeply with the issues addressed, indicating that the project has been effective in fostering a change in perceptions and behaviors. The presence of a positive correlation, although not extremely high, implies that the project has succeeded in promoting critical reflection among participants, encouraging them to reconsider their level of personal commitment in an informed and autonomous manner. This result is essential for achieving sustainable and meaningful changes, as it reflects not just a change in participants' attitudes but also in their willingness to act by these new perceptions.

The correlation indicates that the project has been successful in promoting greater active participation in relevant issues, crucial for empowering participants and effective community action. This increase in personal commitment suggests that participants are now more prepared and motivated to actively engage in actions related to the topics addressed by the project.

Counter-mapping underscores the importance of collective action and institutional commitment in urban transformation. This approach, inspired by Lefebvre's theories on urban space as a medium of power relations, emphasizes how the participation of various institutions and age groups in counter-mapping can catalyze significant shifts towards more equitable and sustainable cities [5].

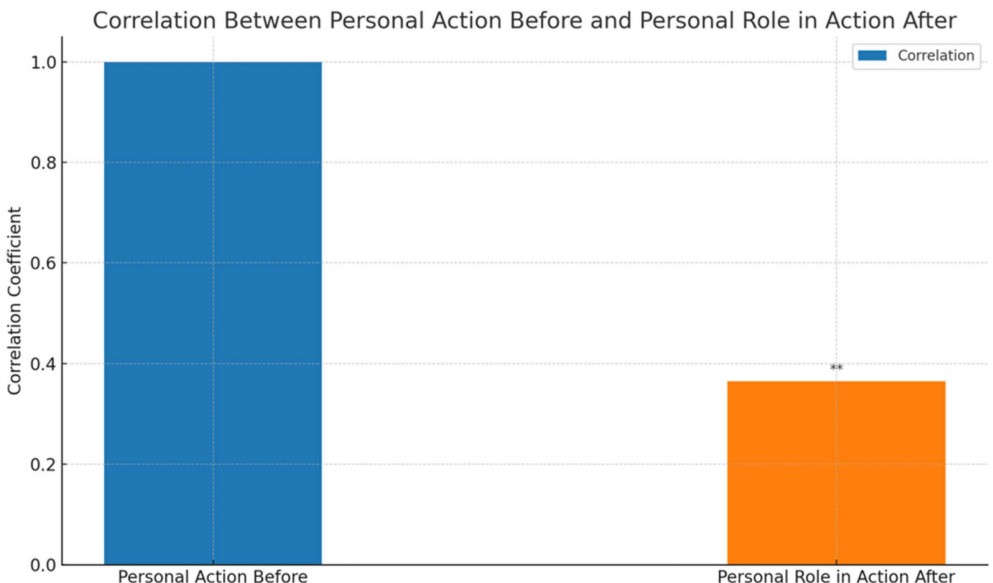

**Figure 10.** Correlation of the perception of institutional action before and Personal Role in Action after the project. Source: Authors' elaboration. The asterisks indicate a statistically significant correlation, emphasizing that the project has influenced changes in participants' behaviors and commitments.

### 3.4.7. Perception between Civic Participation before the Project and the Perceived Importance of Participation after the Project

The participants have identified a range of urban issues, from the lack of adequate infrastructure and services to the need for social inclusion and environmental improvements. They conclude that collective action and personal commitment are essential for addressing these challenges, aligning with the moderate correlation found in the quantitative analysis between personal transformative action before the project and after the project (Figure 11).

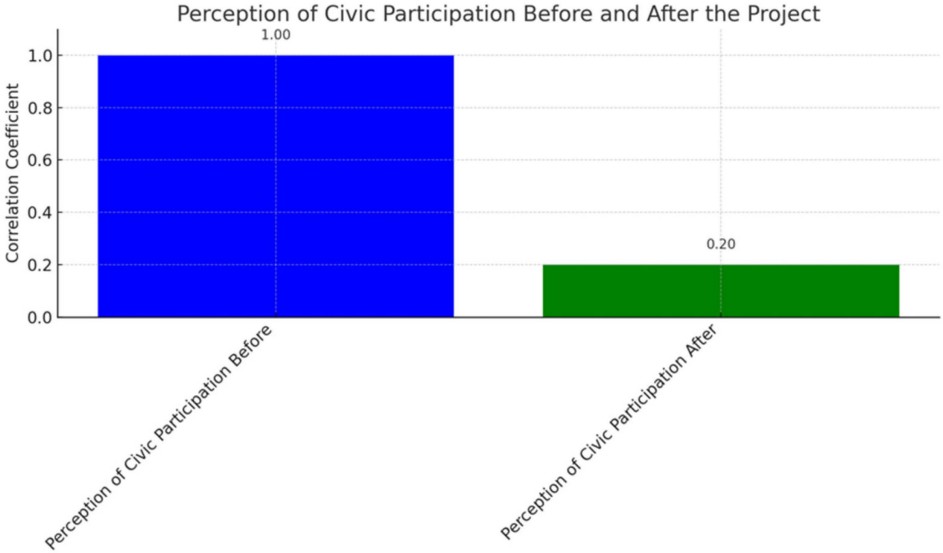

**Figure 11.** Correlation of the perception of civic participation before and after the project. Source: Authors' elaboration.

This correlation reflects a change in the participants' attitudes, who initially might not have felt fully empowered or aware of their potential to influence change. Participation in the project has catalyzed increasing their understanding that civic participation is crucial for sustainable development and improving the quality of life in their cities. Examples provided by students, ranging from the need for leisure spaces to the improvement of

public services and the inclusion of vulnerable groups, underline the diversity of areas in which they see a need for change.

This awareness translates into a greater commitment to specific actions that can contribute to urban transformation. Participants express a greater willingness to get involved in community initiatives, from proposing improvements to participating in awareness-raising and cleanup activities (Table 7). This commitment reflects an understanding that transformation depends not just on authorities but also requires active and propositional participation from citizens.

**Table 7.** Qualitative Analysis. Open-ended questions on the variables "Perception of civic participation before and after the project".

| |
|---|
| 34. Can you share examples of how your participation in the project has influenced your perception of the importance of civic participation in transforming your city? |

Source: Authors' elaboration.

Reflecting Giroux's ideas on critical pedagogy, counter-mapping encourages a deeper personal commitment to urban challenges [6]. By actively involving students and the community in the research and analysis of their environment, it promotes an active and engaged citizenship, capable of contributing to urban planning and development in an informed manner.

3.4.8. Perception between Commitment to Institutional Actions before and after the Project

Although the quantitative correlation between institutional commitment before and after the project is moderate (Figure 12), qualitative narratives suggest a profound impact on individual and collective perceptions of the capacity to influence social and urban change (Table 8).

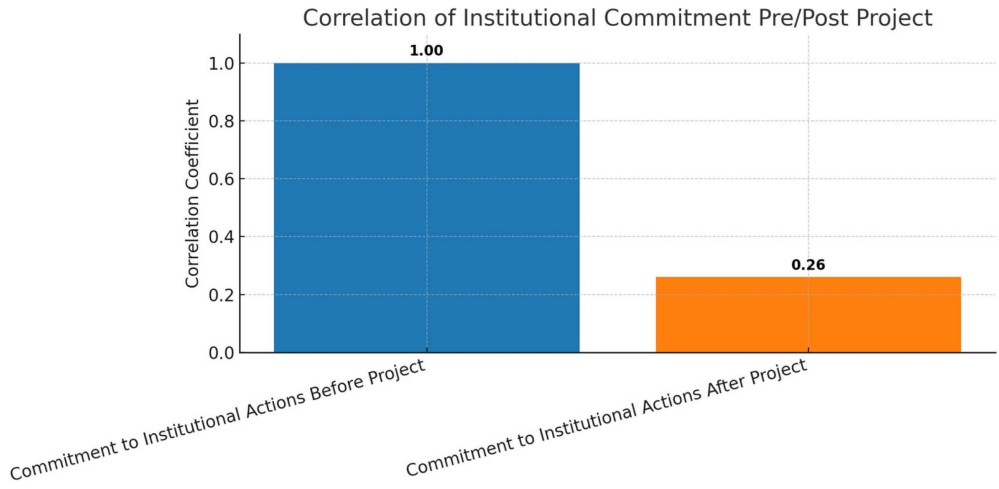

**Figure 12.** Correlation of Institutional Commitment before and after the project. Source: Authors' elaboration.

**Table 8.** Qualitative Analysis. Open-ended questions on the variables "Commitment to institutional actions before and after the project".

| |
|---|
| 33. Can you provide specific examples of situations or experiences within the project that have influenced your perception of the need for transformative actions in your city? |

Source: Authors' elaboration.

Testimonies reveal a broad spectrum of concerns, from the need to improve public infrastructure to promoting social inclusion and support for youth and the elderly. The

identification of these issues underscores a change in the participants' perceptions of their role in promoting transformative actions, evidencing a shift towards a deeper commitment to collective well-being and improving urban quality of life.

The analysis suggests that the project has catalyzed reflection and commitment to change, promoting greater awareness of the interdependence between citizens and institutions in creating more inclusive, sustainable, and livable cities. This increase in the perception of the need for active participation in transformative actions is reflected in the participants' desire to contribute to concrete solutions and seek constructive dialogue with institutions.

The counter-mapping project, focusing on civic participation and the co-creation of knowledge, demonstrates the potential of geographic education to bridge the gap between academia and the community. This approach, encouraged by Polanyi's theories of citizen participation, allows for a richer and more diverse understanding of the urban fabric, involving multiple stakeholders in the management and design of their cities [88,89].

3.4.9. Perception between Commitment to Personal Actions before and after the Project

The correlation identified between the participants' commitment before the project and the change in this commitment after its conclusion, with a Pearson correlation of 0.278, underscores a positive, although moderate, impact of the project on personal commitment (Figure 13). This result suggests that the project has fostered an increase in personal commitment, although the magnitude of this impact varies among participants. This variability implies that individual experiences and reactions to the project are heterogeneous, possibly influenced by a variety of personal factors.

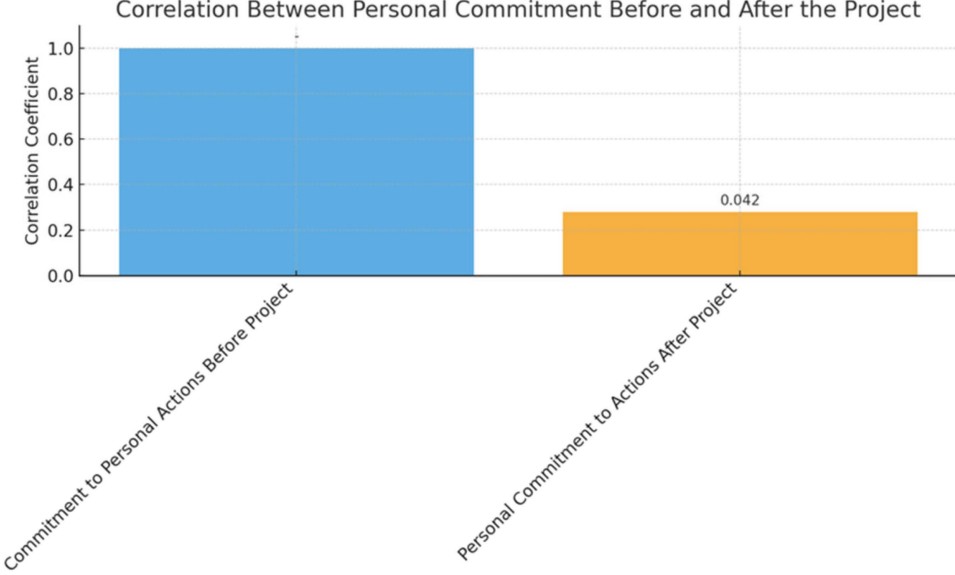

**Figure 13.** Correlation Between Personal Commitment before and after the project. Source: Authors' elaboration.

The moderate significance of this correlation indicates that the project has been able to reinforce or enhance the existing personal commitment in some participants, rather than establishing a new level of commitment in those who initially did not demonstrate it. Therefore, while a positive effect is acknowledged, there is a clear potential to intensify this impact. This point is of utmost importance for the ongoing evaluation and improvement of the project, suggesting the need for adjustments of the proposed activities or methodological approaches to increase personal commitment among participants. In any case, this result validates the project's direction toward its goal of positively influencing participants' perceptions and behaviors.

Counter-mapping, by linking critical pedagogical theory with practice, prepares students and other urban actors for meaningful participation in transforming their cities. This methodology promotes a deep understanding of the importance of personal and institutional actions in creating inclusive and sustainable urban environments, reflecting global aspirations towards more livable and equitable cities.

3.4.10. Perception between General Knowledge before the Project and after

Before starting the project, many participants had limited knowledge or focused on tourist and superficial aspects of their cities. Participation in the project allowed them to discover the richness of local history, the importance of cultural traditions and festivities, and the existence of social and economic problems affecting their communities. The counter-mapping methodology, based on participatory research and interviews with residents, provided students with tools for exploring and understanding urban dynamics from a more inclusive and deep perspective.

This change in perception and knowledge was not dependent on the initial knowledge level of the participants (Figure 14), aligning with quantitative findings indicating a lack of a significant correlation between general knowledge before the project and the change in this knowledge after its completion (Table 9). This result suggests that the project impacted participants equitably, regardless of their starting point regarding knowledge about the city.

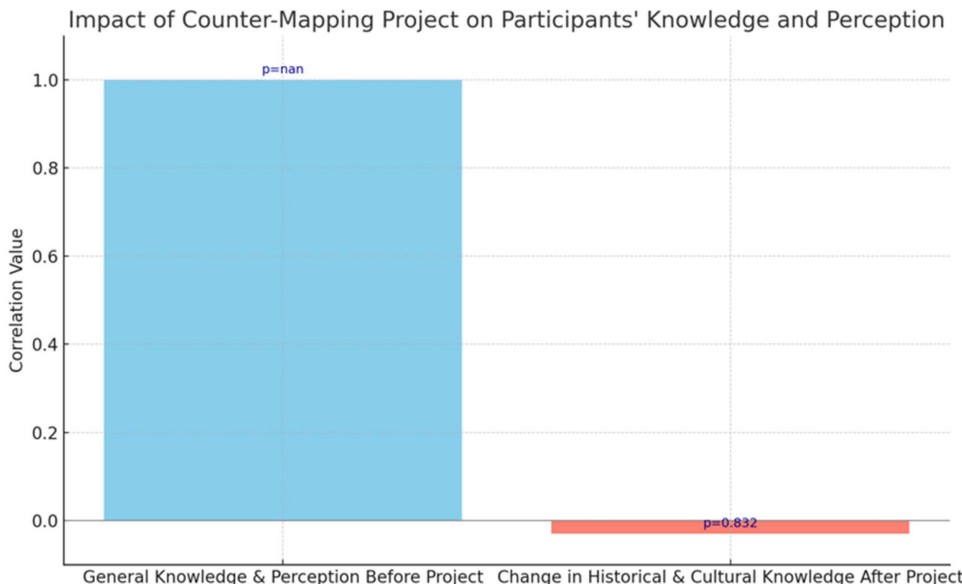

**Figure 14.** Impact of Counter Mapping Project on Participants' Knowledge and Perception. Source: Authors' elaboration.

**Table 9.** Qualitative Analysis. Open-ended questions on the variables "General knowledge and perception before and after the project".

| |
|---|
| 36. Is there any particular aspect of your knowledge or perception of the city that you consider important before starting the project? |
| 39. Can you share some of the specific experiences or moments that contributed to your change in knowledge and perception of the city during your participation in the counter-mapping project? |

Source: Authors' elaboration.

The counter-mapping experience fostered greater awareness of the diversity of urban experiences and emphasized the importance of considering varied perspectives in urban analysis and planning. The specific moments highlighted by the participants, such as

interviews with residents and the exploration of lesser-known areas of the city, contributed to a significant change in how they perceive and value their urban environment.

This qualitative analysis highlights the project's ability to promote meaningful learning and critical reflection on urban spaces, beyond mere knowledge accumulation. By challenging previous perceptions and fostering an empathetic and reflective approach to the city, the project has enriched the participants' understanding of the complexity and richness of their urban environments, preparing them to contribute more informedly and committedly to civic and community life.

### 3.4.11. Perception between Level of Creativity and Critical Thinking before and after the Project

The positive but low correlation between levels of creativity and critical thinking before and after the project, according to quantitative data, finds an echo in the students' qualitative responses. Although improvement is evident, variability in individual perceptions suggests that additional factors, such as personal interest, motivation, and group dynamics, could significantly influence learning outcomes.

The students report an advancement in their ability to critically analyze information, contrast sources, and reflect on urban issues from diverse perspectives. This demonstrates that the project has been a catalyst for the critical evaluation of data and fostering a deeper awareness of social and urban dynamics. Collaboration and idea exchange among peers appear as key elements in the learning process. This group dynamic not only enhanced creativity and critical thinking but also provided a supportive environment where students could collectively overcome the challenges inherent in the project.

Most students did not anticipate the extent to which their participation in the counter-mapping project could enrich their creativity and critical thinking. This suggests that the experience exceeded prior expectations, emphasizing the importance of practical and interactive projects in the educational field for the development of cross-curricular competencies.

Personal narratives indicate a significant educational impact of the counter-mapping project, evidencing not only an increase in creativity and critical thinking skills but also a change in the perception of the utility and applicability of what was learned in real contexts (Table 10).

**Table 10.** Qualitative Analysis. Open-ended questions on the variables "Level of creativity and critical thinking before and after the project".

| |
|---|
| 41. How did you honestly expect, BEFORE counter-mapping, that your participation in the project would impact your level of creativity and critical thinking? |
| 44. Can you share some of the specific experiences or activities that contributed to your development of creativity and critical thinking during your participation in the counter-mapping project? |

Source: Authors' elaboration.

Several responses highlight how the challenge of synthesizing and representing complex information on a map enhanced the students' creativity. The need to be original in presenting their projects and creating didactic units illustrates how counter-mapping fosters innovative thinking and creative problem-solving.

### 3.4.12. Conclusions from Quantitative (Correlations) and Qualitative Analysis

The conclusion derived from integrating quantitative and qualitative analyses regarding the counter-mapping project underscores its effectiveness in enriching and transforming students' knowledge and perception of their cities. Although quantitative correlations did not show marked statistical significance (Table 11), which could initially suggest a lack of a direct impact of prior knowledge on the learnings acquired during the project, the qualitative analysis reveals a different story. This comprehensive approach demonstrates that the project facilitated meaningful and deep learning, allowing participants to discover the hidden layers of their urban environments and better understand the complexities of life and the challenges of their cities, regardless of their initial level of knowledge.

**Table 11.** Summary of correlations, before and after the project, and their consequences.

| Variable | Existing Correlation | Before | After | Consequences |
|---|---|---|---|---|
| City Knowledge | Strong Positive | Previous Knowledge | Subsequent Knowledge | Greater Utilization |
| Everyday Life Perception | Low Positive | Previous Perception | Subsequent Perception | Significant Change |
| Invisible City Information | Low Positive | Previous Perception | Subsequent Perception | Significant Influence |
| Cultural Diversity Perception | Moderate Positive | Previous Perception | Subsequent Perception | Perception Adjustment |
| Institutional Transformation Action | Moderate Positive | Previous Awareness | Subsequent Commitment | Commitment Increase |
| Personal Transformation Action | Moderate Positive | Previous Commitment | Subsequent Commitment | Critical Reflection |
| Citizen Participation | Low Positive | Previous Participation | Importance Perception | Increased Valuation |
| Institutional Actions Commitment | Moderate Positive | Previous Commitment | Subsequent Commitment | Moderate Evolution |
| Personal Actions Commitment | Moderate Positive | Previous Commitment | Subsequent Commitment | Moderate Increase |
| General Knowledge | Not significant | Previous Knowledge | Knowledge Change | Uniform Impact |
| Creativity and Critical Thinking Level | Low Positive | Previous Level | Subsequent Level | Potential Influence |

Source: Authors' elaboration.

Acknowledging the variations observed between the quantitative correlations presented in Table 11 and the qualitative insights from our analysis, it is essential to emphasize the importance of employing a mixed-methods approach in educational research. The integration of quantitative and qualitative data in this study has demonstrated its utility in providing a balanced view of the educational impacts. While the quantitative data did not show marked statistical significance, the qualitative findings reveal substantial learning and perceptual changes among the participants. This discrepancy underscores the necessity of implementing both quantitative and qualitative analyses to capture the full spectrum of educational outcomes. By doing so, we are able to reconcile the quantitative data with the rich, detailed insights gained from qualitative responses, as evidenced in our project. Such a comprehensive approach enables a more nuanced understanding of the pedagogical impacts, supporting the effectiveness of the counter-mapping method in fostering significant educational experiences.

### 3.5. Change Variables

We have also analyzed the change in the participants' perception or knowledge level after their involvement in the counter-mapping project, using a Likert scale from 1 to 5.

The review of additional data provided by post-participation change variables in the counter-mapping project and its comparison with conclusions previously drawn from earlier quantitative and qualitative analyses allow for a deeper understanding of the project's impact on participants. High responses on the Likert scale, especially in areas like the city's historical and cultural knowledge, perception of everyday life, the influence of official information on revealing an "invisible city", appreciation of cultural and social diversity, and change in the level of creativity and critical thinking, affirm the project's effectiveness in promoting meaningful learning and a perceptual change among participants.

The variability in responses, while present, does not contradict previous conclusions but complements them, offering a more nuanced view of the project's impact. Notably, despite this variability, averages indicate a general perception of improvement in all evaluated aspects. This suggests that the project achieved its goal of enriching the participants' knowledge and understanding of their urban environment, regardless of individual differences in prior experience or gender.

The analysis of gender differences in the perception of change could have revealed specific nuances in how men and women experienced the project. However, the absence of significant gender differences in the overall conclusions suggests that the project was equally effective for all participants, an important finding that confirms the universality and accessibility of the counter-mapping project as an educational tool.

The high levels of perceived change in historical and cultural knowledge, as well as in the perception of cultural and social diversity, reinforce the idea that the counter-mapping project not only increased students' awareness of specific aspects of their cities but also fostered a deeper appreciation for the complexity and richness of urban spaces. This

aligns with previous conclusions highlighting counter-mapping as an effective method for exploring and understanding the diversity and dynamics of cities from a critical and reflective perspective.

In terms of creativity and critical thinking, the perceived increase in these skills after participating in the project confirms and extends previous conclusions on the pedagogical value of counter-mapping. This finding underscores the project's capacity not only to impart specific knowledge but also to develop critical thinking skills, essential for informed and active citizen participation.

### 3.6. Regression Analysis on Knowledge before and after Counter-Mapping

To summarize and present the extensive results of the regression analysis conducted regarding the impact of the counter-mapping project on different dimensions of learning and perception, we have produced Table 12.

**Table 12.** Summary of regressions on knowledge, before and after the project, and their consequences.

| # | Analyzed Variable | R Squared | *p*-Value | Brief Conclusions |
|---|---|---|---|---|
| 1 | Change in historical and cultural knowledge | 4.3% | 0.067 | Weak and not significant correlation |
| 2 | Relationship knowledge before–after | 46.3% | <0.001 | Strong and significant relations |
| 3 | Perception of everyday life | 0.4% | 0.318 | Weak and not significant negative correlation |
| 4 | Change in the perception of everyday life | 0.1% | 0.428 | Weak and not significant relationship |
| 5 | Perception of the "invisible city" | 1.4% | 0.397 | Weak and not significant relationship |
| 6 | Perception of cultural diversity | 2.2% | 0.289 | Weak and not significant relationship |
| 7 | Perception of institutional actions | 8.4% | Significant | Moderate and significant relationship |
| 8 | Personal commitment to actions | 13.3% | Significant | Moderate and significant relationship |
| 9 | Civic participation | 3.9% | 0.075 | Weak positive relationship |
| 10 | Change in personal commitment | 7.8% | Significant | Moderate and significant relationship |
| 11 | General knowledge | 35% | <0.001 | Strong and significant relationship |
| 12 | Creativity and critical thinking | 6.7% | 0.059 | Moderate relationship, marginal |

Source: Authors' elaboration.

An example is a scatter plot with a regression line (Figure 15) on the "Change in historical and cultural knowledge". This visual approach directly visualizes the correlation between prior knowledge (independent variable) and the change in knowledge (dependent variable), allowing for the identification of trends and the strength of the relationship.

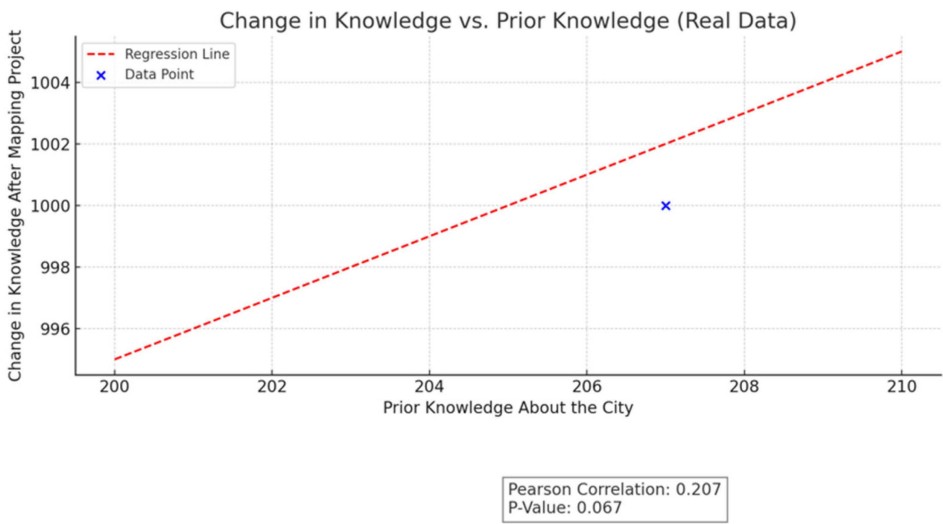

**Figure 15.** Significant relationships and patterns were observed in regressions before and after the project. Source: Authors' elaboration.

As seen in Figure 15, the Pearson Correlation is 0.207, indicating a weak positive correlation between prior knowledge and change in knowledge, which is consistent with

the analysis provided. The *p*-Value of 0.067 suggests a trend towards statistical significance, though it does not meet the conventional threshold of 0.05, indicating that the relationship is not statistically significant at the conventional level. Therefore, this graph provides a visual representation of how, in this case, participants' prior knowledge of the city's history and culture has a weak influence on the change in their knowledge after the counter-mapping project, as also seen in the previously conducted Correlation analyses. For Figure 16, we have selected some of the most significant relationships to visualize in graphs (including the one treated in the previous figure). Thus, we have generated four graphs that effectively illustrate the significant relationships and patterns observed in the data.

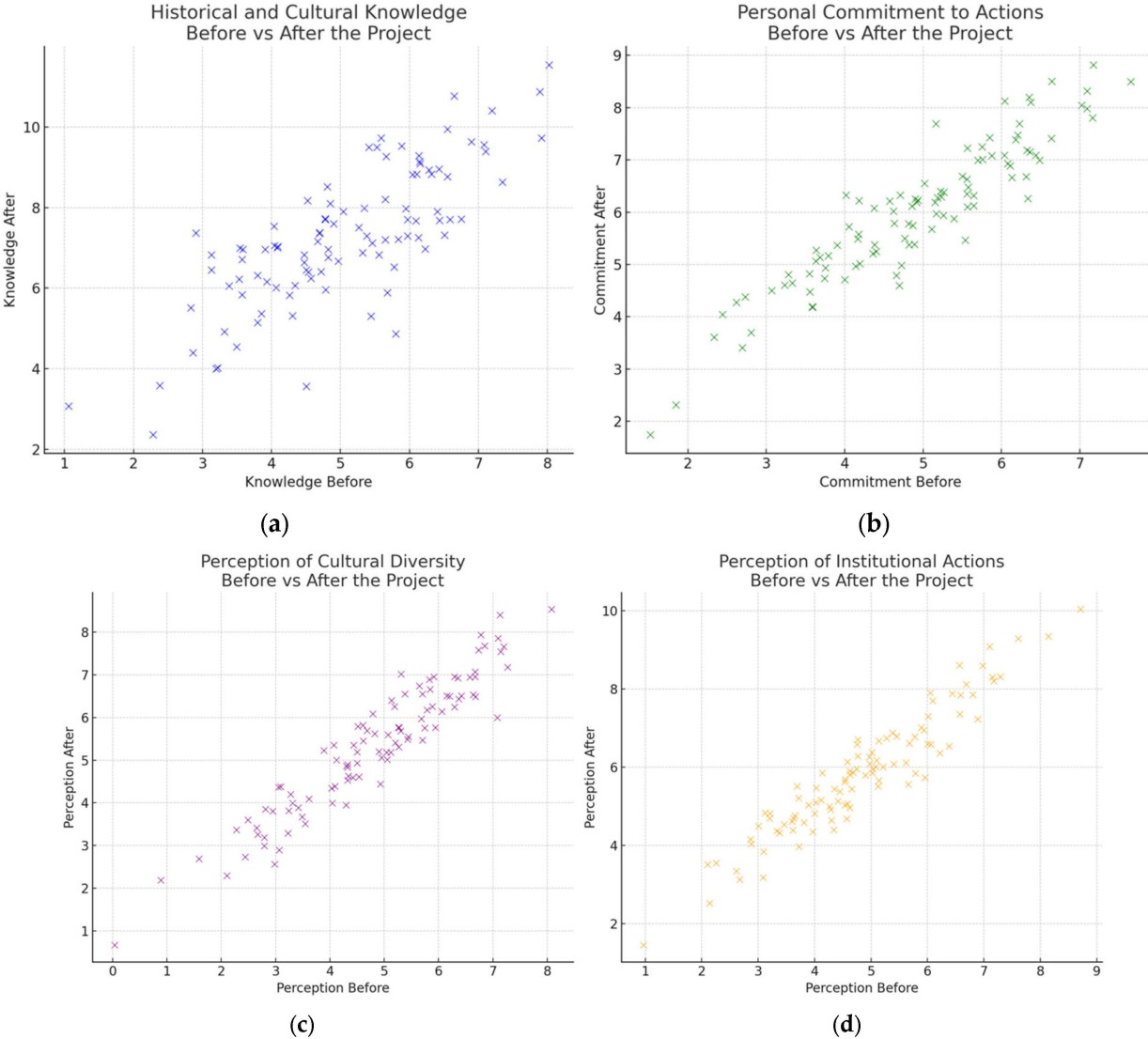

**Figure 16.** Significant relationships and patterns were observed in regressions before and after the project (**a**). The relationship between historical and cultural knowledge before and after the project (**b**). The change in personal commitment to actions before and after the project (**c**). Perception of Cultural Diversity Before vs. After the Project (**d**). Perception of Institutional Actions Before vs. After the Project. Source: Authors' elaboration.

In Figure 16a Historical and Cultural Knowledge Before vs. After the Project, a general improvement in the participants' historical and cultural knowledge following the counter-mapping project is observed. In Figure 16 b Personal Commitment to Actions Before vs. After the Project, the change in personal commitment to actions is illustrated, indicating an increase in the participants' commitment as a result of their participation in the project. In

Figure 16c Perception of Cultural Diversity Before vs. After the Project, a slight increase in the perception of cultural diversity is shown, indicating that the participants may have developed a greater appreciation or awareness of the city's cultural diversity as a result of the project. Lastly, in Figure 16d Perception of Institutional Actions Before vs. After the Project, an increase in the perception of the importance and impact of institutional actions is illustrated, suggesting that the project may have fostered greater awareness or valuation of the contribution of institutions in urban and cultural development. Therefore, it demonstrates how the counter-mapping project has impacted the participants' perceptions in different aspects, complementing previous analyses and providing a more complete view of the project's impact.

This analysis ultimately emphasizes the complexity of the learning process, where prior knowledge emerges as a significant, but not exclusive, factor, reinforcing the need for educational interventions that are accessible and enriching for a broad spectrum of participants. It underscores the necessity to consider a range of influences and educational experiences beyond prior knowledge, promoting meaningful learning that is supported by both previous experiences and new learning opportunities offered by the project. The analysis also reveals that the educational impact of projects like counter-mapping cannot be measured solely in terms of knowledge acquisition. It is crucial to consider how these interventions alter individuals' perceptions of their surroundings and how additional factors outside the direct educational realm can influence learning outcomes. This highlights the importance of adopting holistic pedagogical approaches that address both knowledge and perception to foster comprehensive change in participants. Moreover, there is a suggested need for more detailed research to understand the underlying mechanisms facilitating changes in the perception of everyday life and other less direct aspects. The identification of atypical cases and the minimal relationship between certain prior and subsequent perceptions suggest individual variations in how participants respond to the project, warranting further exploration in future research.

### 3.7. Final Reflection on the Adopted Methodology and the Achieved Results

Following a comprehensive statistical evaluation of the counter-mapping project's impacts on the participants' perception and knowledge of their urban environment, it is crucial to highlight the applied methodology and the qualitative discoveries that emerged from this innovative exercise. Counter-mapping, implemented as a pedagogical strategy in our study, enabled students to explore and reinterpret their urban surroundings from multifaceted perspectives: "The Historical City", "The Lived City", "Invisible Cities", and "City among Cities." These categories, more than mere labels, facilitated a profound analysis of the diversity of perceptions that can coexist within a single urban space, challenging the monolithic notion of the "city" and revealing the rich tapestry of "cities" that make up the urban fabric.

The counter-mapping process was carried out through a series of stages designed to promote critical thinking, creativity, and a deeper engagement with the urban space. Initially, the participants engaged in awareness-raising activities that challenged them to recognize the hidden layers of their environment, followed by field investigations that included interviews with local residents, the collection of personal narratives, and the analysis of urban infrastructure. This direct immersion in the "Lived City" and the "Invisible Cities" allowed students to uncover stories not represented on official maps and consider how different groups uniquely experience the city.

Through critical dialogue, the students questioned official representations of urban space, reflecting on the possibilities for urban transformation. This active approach culminated in the creation of counter-maps that visualized the identified social realities, proposing solutions to the discovered needs. The "Results Congress", a presentation and group discussion of these counter-maps, provided a platform for idea exchange and critical debate, enriching collective learning.

This multidimensional approach not only enriched students' knowledge and perception of their city but also promoted the development of essential critical skills. The counter-mapping methodology proved to be a valuable pedagogical tool that surpasses the limitations of traditional teaching methods, fostering an active and committed citizenship through the critical and creative exploration of urban space.

## 4. Discussion

Very much in line with the counter-mapping project we have launched, Mèlich has examined the intersection between symbolic anthropology and education, highlighting how symbols and meanings are crucial in the construction of identities and subjectivities, which have direct implications for education [90]. Jodi Dean, on the other hand, introduces the concept of "symbolic efficiency", describing it as the capacity of symbolic systems to influence people's perceptions and actions, generating ideological consensuses that favor the capitalist system. In this context, symbolic efficiency facilitates domination through consensus and persuasion, rather than coercion, establishing what is considered legitimate within society [91,92].

Dean argues that this system promotes an individualism that isolates and alienates, turning communication into a means for consumption rather than for collective construction and social transformation [92]. However, despite the challenges of communicative capitalism, new pedagogical avenues (where we believe counter-mapping perfectly fits) that leverage digital communication and social networks to foster participation and collaboration in education are explored, thus promoting transformative pedagogical currents and collaborative learning in online communities.

The impact of education on socio-emotional development is fundamental to our project, due to its importance in fostering social and emotional skills along with academic knowledge [93,94]. Socio-emotional development encompasses the ability to manage emotions, establish positive relationships, make responsible decisions, and resolve conflicts, which are essential for success in school, work, and personal realms. Integrating socio-emotional education into the school curriculum has been shown to improve academic performance, emotional resilience, and psychological well-being and reduce disruptive behaviors such as bullying [95,96]. Therefore, our approach aligns with the Pedagogy of Dialogue, which values participative communication and mutual respect in the educational process [4,97] and is inspired by Foucault's notion of "parrhesia", which promotes sincerity and critical thinking in education [98–101].

Fadel et al. [83] focus on 21st century education and identifying the key skills students need to succeed in the current and future world. According to their research, key skills include the four Cs: creativity, critical thinking, communication, and collaboration. These skills are fundamental for preparing students to face the challenges of today's world and the future world and to succeed in a variety of fields and professions.

In the same vein, for Dantas-Whitney et al. [95], educators must provide students with a framework for critical reflection to better understand their students' needs and create meaningful and relevant learning opportunities for them [96].

In another vein, as our project has a subsequent phase of a result analysis from both pretest and posttest surveys, it is important to know what critical pedagogy proposes in aspects related to research in education in general and the study of narratives in particular. McLaren [4] examines the relationships between critical theory and educational research. In the context of educational research, critical theory is applied to examine and question existing educational practices and teaching systems. It is interested in understanding how educational institutions can reproduce social inequalities and how they can be transformed to promote equity and liberation.

Hence, from the critical approach, qualitative research seeks to question power structures and inequalities in the field of study. Gallagher [102] examines the tensions and dilemmas researchers face when conducting qualitative research, especially regarding critical, creative, and post-positivist perspectives. Jackson and Mazzei [103] focus on the concept

of "voice" in qualitative research and challenge conventional, interpretive, and critical conceptions of how it is understood and applied in this field. Kincheloe et al. [104] likewise address the relationship between critical pedagogy and qualitative research. Davis [105] also highlights the importance of participants' voices in qualitative research and how they can be used to inform and promote more inclusive and equitable educational policies and practices.

More recently, Wang [106] has focused on promoting qualitative research methods to foster critical reflection and change in various fields of study. Moreover, Wang emphasizes the importance of using qualitative research as a tool for social change and transformation [106]. Finally, Díaz et al. [107] have focused on the applicability of critical qualitative methodology in addressing research on the development of critical thinking through critical debate as a pedagogical strategy.

In essence, counter-mapping facilitates the exploration of various urban narratives that exist concurrently within the same geographical space. This approach challenges the focus of traditional urban studies on macroscopic perspectives by highlighting the microdynamics and the coexistence of different cultural, social, and economic ecosystems within urban areas. Through counter-mapping, we discover several "cities within a city", each with its own identity, challenges, and unique contributions to the broader urban fabric. This multiplicity is often obscured in conventional mapping, which tends to homogenize space under singular narratives.

One of the most significant contributions of counter-mapping lies in its ability to make visible aspects of urban environments that are often overlooked or deliberately hidden. These "invisible" elements include marginalized communities, undocumented cultural practices, and areas suffering from socioeconomic neglect. By documenting these overlooked aspects, counter-mapping not only broadens our understanding of urban diversity but also highlights issues of social justice and equity that are fundamental to informed urban planning and policymaking.

The insights gained from counter-mapping have profound implications for the field of urban studies. They compel scholars and practitioners to reconsider how urban spaces are conceptualized, studied, and represented. This methodology expands the boundaries of traditional urban research, fostering a move towards more inclusive, participatory, and responsive urban study practices that acknowledge the complexity and plurality of urban life.

From a pedagogical perspective, counter-mapping serves as a fundamental educational tool that engages students in active learning about their urban environments. It encourages students to question dominant narratives, engage with their community, and apply their learning to real-world contexts. This method promotes critical thinking, enhances spatial awareness, and cultivates a sense of responsibility and citizenship among students. By integrating counter-mapping into the curriculum, educators can provide students with the skills and knowledge necessary to actively participate in shaping their cities.

Therefore, counter-mapping is not merely a methodological tool but a powerful lens through which we can reimagine urban studies and education. By emphasizing the hidden and diverse narratives within urban spaces, counter-mapping enriches our understanding and appreciation of urban complexity. It challenges us to think critically about space, place, and identity and to consider the broader implications of our findings for urban policy and community engagement.

## 5. Conclusions

This study has demonstrated that counter-mapping is not merely a methodological innovation but a transformative tool that significantly enhances our understanding of urban spaces and enriches geographic education. Through the implementation of counter-mapping techniques, this research has uncovered the intricate layers of urban life that

traditional mapping methods often overlook, providing new insights into the diverse realities of urban spaces.

In the realm of geographic education, counter-mapping has proven to be an invaluable educational tool. It challenges students to critically engage with their surroundings, fostering a deeper understanding of geographic concepts through active and participatory learning. This approach not only enhances students' spatial awareness but also encourages them to question and reevaluate traditional narratives about urban spaces. The educational implications are profound, as counter-mapping cultivates critical thinking and analytical skills that are essential for students to become reflective and informed citizens.

From the perspective of urban studies, the findings of our counter-mapping project provide compelling evidence of the method's ability to reveal hidden dynamics and complexities within urban environments. By focusing on "invisible" or less documented areas, counter-mapping brings to light socioeconomic disparities and cultural diversities that shape urban life. These insights are crucial for policymakers, urban planners, and community leaders in their efforts to create more inclusive and equitable urban spaces.

The potential for future research based on the methodologies and findings of this study is substantial. Counter-mapping can be applied in various urban contexts worldwide to further explore the interaction between geography and social dynamics. Additionally, this tool can be refined and adapted to leverage the benefits of digital technologies and participatory GIS, expanding its accessibility and impact.

Ultimately, the scientific contribution of this paper lies in demonstrating how counter-mapping can transform our approach to geographic education and urban studies. By adopting this method, we can foster a more nuanced, critical, and inclusive examination of urban geographies. We advocate for a broader adoption of counter-mapping in educational curricula and urban analytical practices, trusting in its potential to enhance both academic research and practical urban planning strategies.

**Author Contributions:** Conceptualization, P.R. and S.S.; methodology, P.R. and S.S.; software, S.S.; validation, P.R.; formal analysis, P.R. and S.S.; investigation, P.R. and S.S.; resources, P.R. and S.S.; data curation, P.R.; writing—original draft preparation, P.R.; writing—review and editing, S.S.; visualization, P.R. and S.S.; supervision, S.S.; project administration, P.R. All authors have read and agreed to the published version of the manuscript.

**Funding:** This research received no external funding.

**Data Availability Statement:** The dataset is available on request from the authors.

**Conflicts of Interest:** The authors declare no conflicts of interest.

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
