# Peer review of "Counter-Mapping in Geographic Education: A Novel Approach to Understanding Urban and Cultural Dynamics in Cities"

_heritage, doi:10.3390/heritage7050120_

Round 1
Reviewer 1 Report
Comments and Suggestions for Authors
The paper presents a very interesting topic that is especially relevant in today's time of dominance of manipulative media techniques that influence learning and adoption of attitudes, ideas and critical opinions.
The first question that arises concerns the title of the paper: "Reimagining Urban Heritage: Counter-Mapping as a Disruptive Tool in the Transformation of Sustainable and Touristic Cities" - Why is the epithet "disruptive" used for counter-mapping in the title? This implies a negative connotation in terms of the technique itself that is analyzed in the paper. In my opinion, it is enough to emphasize that counter-mapping is a new and opportune method for understanding of urban and cultural environment.
Considering that the paper concerns education, it would be desirable if this term was found in the title of the paper itself, regardless of the fact that the term “Geographic education” is specified in the keywords.
Also, the title of the paper mentions "sustainable and touristic cities", but in the results of the paper (Section 3. Results) there is no research that includes the concepts of sustainability in cities and touristic (or rather cultural tourism) issues. The title seems problematic in terms of these issues and it is necessary to either revise the title or deal more closely with the topic of counter-mapping in terms of sustainability and tourism.
The same applies to the introductory chapter (1. Introduction) - there is no question of the relation of the chosen counter-mapping method to concepts such as sustainability and tourism - does the paper deal with the topic of sustainable cities and tourism?
The Introduction chapter draws the most attention to critical pedagogy and analyzes the importance of different educational approaches that are relevant to the counter-mapping project. However, the central topic of counter-mapping is mentioned in line the 133, it would be advisable to point out at the very beginning why the counter-mapping was chosen and how this method is related to the educational project conducted, what is the advantage of this method compared to other methods of critical pedagogy, and what benefits does this method provide.
The introductory chapter is extremely extensive and includes 99 reference sources. You should think about shortening the introduction, where special emphasis will be placed only on the importance and contribution of the main subject of the work that is counter-mapping, leaving the most relevant opinions such as Lefebvre, Rogers, Lyotard, etc., and moving the remaining reference sources and theoretical ideas to the chapter 2. Materials and Methods.
After the introduction, the "General Objectives" and "Specific Objectives" are listed - this kind of enumeration by points is more suitable for master's and doctoral theses, but it is certainly useful for explaining the research methodology, so you should consider moving this part from lines 151-182 to chapter 2. Materials and Methods, and to single out only the most relevant goals.
In line 222 translate into English the title „2.6. Método de contra mapeo“.
In the Results chapter, a very thorough analysis related to the experiences of participants before and after participation in counter-mapping project is presented. The paper presents an interesting correlation between what is defined in the paper as: "The Historical City", "The Lived City", "Invisible Cities", "City Among Cities" - because it underlines the diversity of perception of the city environment itself, which can be viewed as one entity - "city" and collective entities "cities", however, only statistical analysis is available to us, and we do not see what counter-mapping looked like, through which a number of parameters from general knowledge and perception of the city, creativity and critical thinking were examined.
In the very long analysis given in the 3. Results chapter, however, we don’t find out anything about the form in which the counter-mapping project was implemented in the sections related to: "3.4.2. Perception of Everyday Life"; "3.4.3. Influence of the Information from the Invisible City". We discover that the examined students had a low level of knowledge and perception related to every day life in the observed area of ​​the city, cultural and social differences, social and cultural relations and interactions, about the existence of "cities within a city", and that after the project they broadened, improved and expanded their knowledge. However, the research is focused exclusively on the statistical analysis of the level of knowledge of students before and after participation in the project, but does not provide us with any explanatory information about the way in which the counter-mapping project worked.
Through the results, which are reduced exclusively to a very uniform statistical analysis, the connection of these results with reference sources from pedagogy and critical social theory, which is presented in the introductin chapter of the paper, is not visible. Through the following chapters (3.4.4. - 3.4.9) through statistical comparisons of the results before and after the counter-mapping project, we discover that after participating in the project, the participants came to the formation of attitudes that are more realistic and socially engaged, which included also taking into account the influence and engagement of institutions. The question that arises is what are the benefits of such a project and in what way can they participate in it not only students who deal with city planning and geography issues, but also other actors, especially the local community, different age and social groups, and the state and NGO institutions - what are the advantages of counter-mapping on a broader level from community life centered to the global city.
The paper presents the idea that counter-mapping is a very suitable tool in the processes of development and improvement of the city studies, especially because this method improves our perception of the environment, and it is considered relevant to more precisely explain the essence of the counter-mapping process, especially when talk about changes, diversity and dynamics of cities. It seems to me that for the scientific contribution of this paper it is very important to explain in the paper the mentioned concepts through which critical learning is carried out, such as: "perception of every day life", "invisible city", "cities within a city", "creativity and critical thinking", etc., rather than just offering comparative statistical data on the respondents' perception before and after participating in the project.
The chapter 4. Discussion should be connected with the results of the research (3. Results). It is useful that the authors refer to theoretical sources and studies that deal with educational issues, but there is no clear connection between chapter 4. Discussion and chapter 3. Results.
The biggest objection to the paper as a whole is that the title itself suggests the importance of counter-mapping on sustainable and touristic issues of cities, and the paper did not say anything about it. The paper presents a statistical study that deals only with the final results of the project, while the important elements that are key to understanding the importance and contribution of the counter-mapping method are omitted, in the Introduction and Discussion the theoretical sources point to a very interesting project, but we do not see anything of that in the results - only statistical analyses related to the group of student respondents. Therefore, it is necessary to strengthen the section devoted to the Discussion and Conclusion and to summarize and connect the section 3. Results with theoretical references relevant to this project.
Author Response
Dear Reviewer,
We greatly appreciate your insightful comments, which have significantly contributed to the enhancement of our manuscript. Attached, please find our responses and explanations regarding the modifications made, along with the precise locations of these changes in the revised manuscript submitted.
Thank you again for your valuable feedback.
Best regards,
Dr. Seila Soler
Dr. Pablo Rosser

Reviewer 2 Report
Comments and Suggestions for Authors
The work presented is congratulated and commented on:
1. In the introduction it is essential to contextualize the case study or the universe of study, since there is no reference to it and it is necessary to understand the contribution in that context.
2. It is necessary to know the scope of the research project to which the work refers recurrently.
3. It is necessary to present the objectives of the work in an articulated manner, i.e., the list format is not appropriate for this type of text and needs to be adjusted. It could even be synthesized in the introduction to be developed in detail in the methodology to demonstrate the application of techniques and instruments.
4. It is required to report in detail the applicable legal considerations for the use of personal information and data. It is not sufficient to state the application of ethical principles.
5. In the methodology, it is necessary to state the period or periods of time during which the research was carried out. This is fundamental for the presentation of results in the sense of evidencing significant improvement of knowledge.
6. In the methodology it is necessary to present at least the areas in which the research is conducted, i.e., the general contents measured in the participants.
7. Check that the figures are legible in all their content.
8. Relocate or move the figures to avoid jumps in the pages and the interruption of the continuity of the contents.
9. Review the labeling of components in the figures, example, figure 4. In addition, they should be unified in terms of their presentation, example, figure 8.
10. Check that the arguments issued have an objective support and are not only limited to subjective descriptions of the authors. Example, lines 360-364.
11. In addition to presenting the results separately, it is necessary to present the integrated analysis of the criteria presented. I recommend reviewing the presentation and analysis of statistical mosaics and their contribution to the comprehensive understanding of results that contribute to the same objective. Figure 17 is a good example. This consideration would make it possible to reduce the number of figures, which is exaggerated.
12. Contents such as Table 2 should be relocated to methodology. Revise this comment in relation to number 6 of this list.
13. Improve the presentation of Table 12, it should not be distributed in 3 different pages. The information does not merit it.
14. Review the wording and possible redundancies in the last paragraph of the results.
15. It is suggested to articulate in a single section Results and Discussion. The discussion seems isolated in relation to the specific findings stated. In turn, what is presented is more oriented towards introduction than discussion.
16. It is recommended to improve the argumentation of the conclusions making effective use of the significant findings; avoid presenting contents as an introduction.
Comments on the Quality of English LanguageGeneral review of the document
Author Response

(The authors gave the same response as above.)

Reviewer 3 Report
Comments and Suggestions for Authors
Counter Mapping is one of the promising research techniques useful for a holistic and multidimensional analysis of problems involving material resources and components of human involvement (cognitive, social and other), which is typical in cases of heritage research and its potential (both social and tourist). In this context the conclusion from the study about the discrepancies in the results obtained using quantitative methods (no statistical differences) and qualitative methods (significant differences in the increase in knowledge and involvement of the respondents) seems to be important. This delivers an argument for the use of triangulation of research methods in the analysis of heritage awareness, heritage tourism and tourism planning. Proving this on the example of a local study, conducted in an exemplary manner, should be considered the greatest advantage of the reviewed text.
Counter Mapping is one of the promising research techniques useful for a holistic and multidimensional analysis of problems involving material resources and components of human involvement (cognitive, social and other), which is typical in cases of heritage research and its potential (both social and tourist). In this context the conclusion from the study about the discrepancies in the results obtained using quantitative methods (no statistical differences) and qualitative methods (significant differences in the increase in knowledge and involvement of the respondents) seems to be important. This delivers an argument for the use of triangulation of research methods in the analysis of heritage awareness, heritage tourism and tourism planning. Proving this on the example of a local study, conducted in an exemplary manner, should be considered the greatest advantage of the reviewed text.
Author Response

(The authors gave the same response as above.)

Reviewer 4 Report
Comments and Suggestions for Authors
1. The authors reviewed relevant theories and literature on critical pedagogy. However, there is only a brief description of the counter-mapping related to this article's primary purpose, and it is recommended to be supplemented.
2. This article focuses on the implementation results of counter-mapping and measures it through pre- and post-test comparisons. However, there is no specific description of the implementation method of counter-mapping.
3. Among the statistical values ​​in Figure 2, two items are after the project is implemented, one item is not marked, and the rest are before the project is implemented. A more detailed explanation of statistics and comparison methods is required.
4. The paired samples in Figure 3 are labeled 1, 3, 4, 5, 6, 7, 8, 9, 10, 11, and 2 is missing. In addition, only the value of paired 3 is negative, and the rest are positive. The reason needs to be stated.
5. Are the statistical methods used for each item in this study consistent? Integrating and using qualitative and quantitative data also needs to be explained in detail.
6. This study failed to produce insightful or original results. It is recommended that the authors review and think again.
Comments on the Quality of English LanguageNo.
Author Response

(The authors gave the same response as above.)

Reviewer 5 Report
Comments and Suggestions for Authors
Inadequate description of sustainability and touristic profile of cities as field of resarch.
Aren't lines 40-42 the repetition of the earlier text part?
Line 222 should be translated to English.
Line 307: -0,1 (Pair 3) is "notable change"? What about Pair 2 (Fig. 3)?
What is significance of the wideness of the columns difference (Fig. 9 and Fig. 10)?
Author Response

(The authors gave the same response as above.)

Round 2
Reviewer 1 Report
Comments and Suggestions for Authors
The authors have greatly improved the article, although there are still certain elements that can be further improved.
Author Response
Dear Reviewer,
We deeply appreciate your invaluable contributions throughout the revision process of our manuscript titled "Reimagining Urban Heritage: Counter-Mapping as a Disruptive Tool in the Transformation of Sustainable and Touristic Cities." Your insightful feedback has been instrumental in enhancing the clarity, depth, and overall quality of our work.
Following your suggestions, we have meticulously revised the structure, content format, and graphics of our manuscript. We have integrated brief paragraphs into larger sections for coherence, refined the typography and sizing in our figures for better legibility, and ensured that no content is repeated or misplaced.
We have also expanded upon the pedagogical design of the counter-mapping method, incorporating a detailed semester-long course outline that vividly illustrates the educational strategy. This enhancement not only clarifies our methodological approach but also reinforces the innovative aspects of our educational framework.
Furthermore, we addressed the integration of qualitative and quantitative methods in our evaluation framework to better correlate our instructional design with assessment methods, ensuring a comprehensive and reliable evaluation of the pedagogical impacts.
Your thorough reviews have undoubtedly guided us to a more structured presentation of our research findings and methodology. We trust that these modifications have successfully addressed your concerns and contributed significantly to the advancement of our manuscript.
Please let us know if further revisions are necessary or if there are additional aspects of the manuscript that could benefit from more refinement. Your ongoing feedback is crucial to achieving the highest standard for our collaborative academic work.
Thank you once again for your meticulous attention and constructive suggestions throughout this revision process.
Warm regards,
Dr. Pablo Rosser
Dr. Seila Soler

Reviewer 2 Report
Comments and Suggestions for Authors
We thank the authors
The authors are thanked for their comments. On this occasion I would like to point out the following in terms of form:
1. Check the writing to present consistent paragraphs. consistent paragraphs. That is to say, there are paragraphs of 3 -4 lines that can contribute to preceding or subsequent contents. contribute to preceding or subsequent contents.
2. Figures present differences in general size, i.e., of the whole figure, but also of the typography of contents and titles. of contents and titles. Some contents are illegible because they are superimposed.
3. Check line 465 and verify that it is not repeated in the rest of the document. of the document.
4. The number of very small tables systematically cuts the reading of the document.
Good luck!
Author Response
Dear Reviewer,
We are immensely grateful for your detailed and constructive feedback on our manuscript, "Reimagining Urban Heritage: Counter-Mapping as a Disruptive Tool in the Transformation of Sustainable and Touristic Cities." Your insights have been crucial in enhancing the presentation and scholarly rigor of our work.
In response to your suggestions, we have made several significant revisions:
1. We improved the consistency and flow of our content by re-evaluating and adjusting the structure of our document. Specific paragraphs were realigned to ensure a more coherent narrative.
2. We addressed the issues regarding figure and typography sizes within our manuscript. Adjustments were made to ensure clarity and legibility, which has significantly improved the visual presentation of our data.
3. We corrected an error with an image caption and ensured that no text was repeated or misplaced throughout the document.
4. To enhance readability, we resized numerous tables that previously disrupted the document's flow. Each table has been standardized to ensure a seamless reading experience.
These modifications not only address your initial concerns but also substantially contribute to the clarity and effectiveness of our manuscript. We sincerely appreciate the role your comprehensive reviews have played in this process.
Please let us know if there are any further refinements you suggest or additional aspects of the manuscript that could benefit from more detailed attention. Your expertise continues to be invaluable to us.
Thank you once again for your guidance and support.
Best regards,
Dr. Seila Soler
Dr. Pablo Rosser

Reviewer 4 Report
Comments and Suggestions for Authors
1. The manuscript's authors have added much information according to my reviewing comments.
2. However, there are three important issues for the manuscript, including:
(1) The pedagogy design, especially how to involve the counter-mapping method, is not exhaustive. For a semester course, the authors should present the design for the teaching units or even the syllabus, which could help the reviewer understand and evaluate innovativeness. At present, the innovativeness of the pedagogy design is not significant.
(2) How to measure teaching effectiveness integrating qualitative and quantitative methods is another important issue. For this issue, the methods for different variables lack consistency. The correlation between assessment methods and instructional design is quite unspecific.
(3) For the last issue, by Table 11 and Table 12, the study consequences of different variables differ. The relationship with the pedagogy design should be reviewed and proposed.
3. An academic article should not be the outcome of a case study. The authors should present innovativeness, regardless of the pedagogy design, assessment methods, and analysis of the research outcome.
Comments on the Quality of English LanguageI don’t have comments on the quality of the English language.
Author Response
Dear Reviewer,
Thank you profoundly for your detailed feedback on our manuscript "Reimagining Urban Heritage: Counter-Mapping as a Disruptive Tool in the Transformation of Sustainable and Touristic Cities." Your insights have been vital in refining and substantially enhancing our work.
Following your recommendations, we have made comprehensive revisions to our manuscript:
1. **Pedagogical Design**: We expanded on the pedagogical design by detailing the semester course structure that employs counter-mapping as its central theme. We now include a full syllabus and descriptions of five dynamic learning stations, which engage students in critical and creative ways to explore urban spaces. This approach enhances their understanding and skills, offering a robust response to your call for a clear delineation of our educational framework.
2. **Assessment Methods**: We addressed your concerns regarding the measurement of teaching effectiveness by integrating a coherent blend of qualitative and quantitative methodologies. This integration ensures that our assessment aligns with the instructional design, enhancing the accuracy and reliability of our pedagogical evaluations. We've incorporated a triangulation method to ensure a comprehensive assessment, providing a strong basis for our innovative instructional approach.
3. **Methodological Consistency**: In response to your observations on the consistency between our assessment methods and pedagogical design, we've clarified the relationship between quantitative and qualitative findings in our study. This approach underscores the utility of a mixed-methods framework in capturing a full spectrum of educational outcomes, thereby supporting the effectiveness of the counter-mapping method.
Each of these revisions has not only addressed your specific concerns but also enriched the manuscript’s academic rigor and clarity. We are confident that these improvements have significantly enhanced the quality and presentation of our research.
We are grateful for the role your thoughtful and constructive critique has played in this process and hope our revisions meet your expectations. Please let us know if further adjustments are needed.
Warm regards,
Dr. Seila Soler
Dr. Pablo Rosser
